# Broad protective vaccination against systemic *Escherichia coli* with autotransporter antigens

Yikun Xing [1,2], Justin R. Clark[1,2], James D. Chang[1,2], Dylan M. Chirman[1,2], Sabrina Green[1,2], Jacob J. Zulk[1,2], Joseph Jelinski[1,2], Kathryn A. Patras[1,3], Anthony W. Maresso [1,2]*

**1** Department of Molecular Virology and Microbiology, Baylor College of Medicine, Houston, Texas, United States of America, **2** TAILΦR Labs, Vaccine Development Group, Baylor College of Medicine, Houston, Texas, United States of America, **3** Alkek Center for Metagenomics and Microbiome Research, Baylor College of Medicine, Houston, Texas, United States of America

* maresso@bcm.edu

**Data Availability Statement:** The alignment data between domains 1 and 2 of SinH and the domains of SinH from solved structures of invasin and intimin have been deposited at Dryad dataset

## Abstract

Extraintestinal pathogenic *Escherichia coli* (ExPEC) is the leading cause of adult life-threatening sepsis and urinary tract infections (UTI). The emergence and spread of multidrug-resistant (MDR) ExPEC strains result in a considerable amount of treatment failure and hospitalization costs, and contribute to the spread of drug resistance amongst the human microbiome. Thus, an effective vaccine against ExPEC would reduce morbidity and mortality and possibly decrease carriage in healthy or diseased populations. A comparative genomic analysis demonstrated a gene encoding an invasin-like protein, termed *sinH*, annotated as an autotransporter protein, shows high prevalence in various invasive ExPEC phylogroups, especially those associated with systemic bacteremia and UTI. Here, we evaluated the protective efficacy and immunogenicity of a recombinant SinH-based vaccine consisting of either domain-3 or domains-1,2, and 3 of the putative extracellular region of surface-localized SinH. Immunization of a murine host with SinH-based antigens elicited significant protection against various strains of the pandemic ExPEC sequence type 131 (ST131) as well as multiple sequence types in two distinct models of infection (colonization and bacteremia). SinH immunization also provided significant protection against ExPEC colonization in the bladder in an acute UTI model. Immunized cohorts produced significantly higher levels of vaccine-specific serum IgG and urinary IgG and IgA, findings consistent with mucosal protection. Collectively, these results demonstrate that autotransporter antigens such as SinH may constitute promising ExPEC phylogroup-specific and sequence-type effective vaccine targets that reduce *E. coli* colonization and virulence.

## Author summary

Extraintestinal pathogenic *Escherichia coli* is the leading cause of adult life-threatening sepsis and urinary tract infections. A vaccine against *E. coli* is essential to both prevent the spread to susceptible hosts and reduce death and disease. Using a comprehensive computational virulome and metagenomics approach, we identified a surface-exposed,

repository with the name "Structure Prediction: SinH full length" and the digital object identifiers (DOI) (https://doi.org/10.5061/dryad.jm63xsjfx). The data package has been cited in the reference and methods section in the manuscript as "Chang J, Xing Y. Structure Prediction: SinH full length [dataset]. 2023 Jan 6 [cited 2023 Jan 22]. Dryad Digital Repository. Available from https://datadryad.org/resource/doi:10.5601/dryad.jm63xsjfx." All other relevant data are within the manuscript and its Supporting Information files.

**Funding:** This work was supported by the National Institute of Allergy and Infectious Diseases (NIAID) of the National Institutes of Health under award numbers U19AI157981 (AWM, KAP) and U19AI144297 (AWM). The funders had no role in study design, data collection and analysis, decision to publish, or preparation of the manuscript.

**Competing interests:** The authors have declared that no competing interests exist.

pathogen-specific autotransporter protein, SinH, as a potential vaccine candidate for *E. coli* infection. The known virulence functions of autotransporters include adhesion, aggregation, and invasion, all critical functions for systemic dissemination, thus highlighting their potential as prophylactic vaccines. We found that vaccination with SinH-based recombinant antigens is sufficient to elicit a broad protective immunity against colonization, bacteremia, and acute urinary tract infection while also lowering the risk of translocation from the intestinal tract. Induction of both systemic and mucosal antibodies likely play a role in protection against infection. The targeting of autotransporters shows promise to combat the increasing global burden caused by multi-drug resistant pathogens, especially against highly pleiotropic bacteria such as *Escherichia coli*.

## Introduction

Extraintestinal pathogenic *E. coli* (ExPEC) is the leading cause of invasive bacteremia and urinary tract infections (UTI), and the second leading cause of neonatal meningitis [1,2]. In human intestinal microbiota, the ExPEC group includes uropathogenic *E. coli* (UPEC), neonatal meningitis *E. coli* (NMEC), and sepsis-associated *E. coli* (SEPEC) [3–6]. ExPEC acquired specific virulence factors that confer them an ability to cause infections at nonintestinal alternative ecological niches, such as the urinary tract, bloodstream, and prostate [7,8]. Clinically, from 1999-to 2014, 6% of all deaths in the U.S. were sepsis-related deaths [9]. ExPEC is still the most common gram-negative organism resulting in severe sepsis, which caused 20% of severe sepsis cases from 1999-to 2008 [10]. In addition, septic patients represent a disproportionately high burden in cost and hospital utilization. In 2013 alone, nearly over 970,000 sepsis cases were admitted annually in the U.S, and sepsis accounted for more than $24 billion in hospital expenses, representing 13% of total U.S. hospital costs [11]. A recent report assessing 204 countries and territories showed antibiotic-resistant pathogenic *E. coli* to be a leading cause of mortality associated with drug resistance (approximately 200,000 deaths attributable to AMR *E. coli* and approximately 800,000 deaths associated with AMR *E. coli* in 2019) [12].

The emergence and increasing prevalence of multidrug-resistance (MDR) ExPEC strains are considered one of the primary drivers behind the global antimicrobial resistance (AMR) crisis [13]. This crisis adds millions of AMR infections annually and a 20-billion-dollar surplus in direct healthcare costs in the United States [14]. A single ExPEC clone, sequence type 131 (ST131), produces an extended-spectrum β-lactamase (ESBL), specifically an enzyme named CTX-M-15, which has fueled the emergence of resistance ExPEC strains globally [15,16]. Additionally, unlike the other non-ST131 ESBL-producing *E. coli* lineages (ST38, ST405, and ST648), ST131 genomes showed unique virulence profiles. For example, the presence of several serine proteases, autotransporters, and UPEC-specific virulence genes were identified exclusively in ST131 isolates [17]. Furthermore, ST131 has an expanded number of virulence genes in contrast to the non-ST131 isolates, which shared similar but varied and lower numbers of virulence genes [7,17]. These advantages possess the ability of ST131 to make a balance between broad colonization, virulence, and antibiotic resistance without a fitness cost, which might serve as the potential driver of ST131 success in causing an enormous number of human infections globally [7,17]. In the U.S., the ST131 clonal group is now considered the most prevalent and extensively dominant antimicrobial-resistant *E. coli* strain overall [18].

An ExPEC-specific vaccine would provide a prophylactic option to reduce mortality associated with severe *E. coli* infections. Indeed, several groups have sought to induce protective immunity against pathogenic *E. coli* with efforts focused on heat-killed inactivated bacteria

vaccines or conjugates of O-antigens to elicit protective immune responses [19–22]. However, due to the failure of heat-killed inactivated bacterial vaccines to prevent uncomplicated UTI and the antigenic heterogeneity of the surface polysaccharide of the *E. coli*, these attempts have had limited further development [19,21–24]. Recent studies indicate that some surface-exposed molecules, such as FimH adhesin (Type I fimbriae adhesin) [25–27], P fimbriae (PapDG complex), Dr fimbriae [26,28–30], iron receptors (FyuA, Hma, lutA, IreA) [31–35], adhesin FdeC [28], or siderophores (iron-chelating compounds) [36] have been shown to induce protective immunity. Despite numerous anti-*E. coli* vaccine studies spanning greater than five decades, no *E. coli* vaccine has been approved by the US FDA. Therefore, there is an urgency to identify a novel, phylogroup-specific, conserved, immunogenic, and protective vaccine against ExPEC.

Building off the work of others [37], we established criteria for the selection of vaccine candidates, including that the antigens be (i) pathogen-specific (able to identify the pathogenic *E.coli* from commensals), (ii) surface-exposed (to be able to be recognized by the immune system), have (iii) high prevalence in clinical isolates (since these strains contain pleiotropic virulence which makes monovalent antigen identification more challenging), (iv) be involved in critical steps in *E. coli* pathogenesis, including either adherence, invasion, translocation, or survival and growth in blood or urine and finally (v) facilitate robust production that is scalable and economic. Our lab has screened a collection of 107 complete *E. coli* genome sequences representing all defined phylogroups and established a manually curated database housing the genetic information of many virulence factors from *E. coli* and other pathogenic bacteria. This allowed for a heat-map like approach that facilitated the identification and frequency of genes and alleles across phylogroup, sequence type, and pathotype. Of the approximate 400 virulence factors assessed from this test group, and scaled for over 20,000 additional genomes in public databases, a gene encoding for a structural invasin-like autotransporter protein [38], termed *sinH*, was observed to be widely distributed in ExPEC pathotypes and ExPEC-associated phylogenetic group B2/D. Of particular interest, this gene is not commonly found in putative commensal *E. coli* or phylogenetic groups associated with systemic dissemination [39]. AlphaFold2 and homology analysis predicted SinH structure to resemble that of invasin of *Y. pseudotuberculosis* or intimin of Enterohemorrhagic *Escherichia coli*, with three Ig-like domains of SinH predicted to be extracellular. *sinH* was also highly prevalent in clinical *E. coli* isolates obtained from pyelonephritis and urosepsis patients [38,40]. Furthermore, a recent paper indicates the *sinH* gene is upregulated during urinary tract infections in women, and the *sinH* mutant strain displayed a significant fitness defect and inflammation reduction in the bladder in the murine model of ascending UTI [38]. Based on the comparative genomic and Alpha-Fold simulated protein structure information and the critical function of this gene for bladder colonization and *E. coli* infection, we believe SinH is a potential vaccine candidate for *E. coli* infection. Here we present data that suggests SinH elicits robust and broad protection from various ExPEC strains and sequence types in multiple murine models of infection.

## Materials and methods

### Ethics statement

All methods performed on mice were conducted in accordance with relevant guidelines and regulations from "The Guide and Care and Use of Laboratory Animals" (National Institute of Health). The Animal Use Protocol number AN-5177 was approved by Baylor College of Medicine's Institutional Animal Care and Use Committee.

## Bacterial strains and culture conditions

The *E. coli* strains used in this study were cultured overnight from a single colony in Lysogeny broth plate (LB; 10 g/l tryptone, 0.5 g/l sodium chloride (NaCl), and 5 g/l yeast extract) at 37˚C after resuscitation from a frozen stock (−80 ˚C, 10% glycerol). ExPEC ST131 strains JJ1886, JJ2050, JJ2528, and JJ2547 were kindly provided by James R. Johnson [41]. Uropathogenic *E. coli* (UPEC) strains UTI89 (O18:K1:H7, ST95) [42] and CFT073 (O6: K2:H1; ATCC 118 #700928, ST73) [43] were kindly provided by Kathryn Patras. *E. coli* strains. W0060 (ST95-like), W0040, W0088, W0116 (ST73-like) were isolated from the blood or feces of hospitalized patients with bacteremia. The number of CFU delivered was calculated by correlating the OD at 600 nm to the number of colonies after plating.

## Plasmid construction

The genes encoding the candidate vaccine antigens were cloned from ExPEC sequence type 131 (ST131) strain JJ1887 genomic DNA (SinH-Ig-like domains-123, encoding the C-terminal passenger Ig-like domains-1,2 and 3 fragments of *sinH*, amino acid residues 337 to 724, hereinafter called SinH-123; SinH-Ig-like domains-3, encoding the C-terminal passenger Ig-like domain-3 fragment of *sinH*, amino acid residues 602 to 724, hereinafter called SinH-3). Both candidate vaccine sequences were sent to the GENEWIZ company (South Plainfield, NJ) for plasmid construction. Both protein domains were cloned into the BamHI and SmaI restriction sites of pGEX-2TK to produce N-terminally glutathione-S-transferase (GST)-tagged fusions (GST-SinH-3, in short SinH-3 in following; GST-SinH-123, in short SinH-123 in following). The resulting constructs were verified by sequencing.

## Vaccine antigens preparation

Both recombinant proteins were produced by *E. coli* BL21(DE3) cultured in Lysogeny broth (LB) to an optical density at 600 nm ($OD_{600}$) of 0.6–0.8. The gene expression was induced with 1mM Isopropyl β-D-1-thiogalactopyranoside (IPTG) (Sigma-Aldrich, St. Louis, MO) and the culture was then incubated overnight at 30˚C. The cells were harvested by centrifugation (10,000 × g for 30 min at 4˚C), and bacterial pellets were resuspended in 1× phosphate-buffered saline (PBS). Bacterial suspensions were lysed by two passages through a French pressure cell press (1500 PSIG) (Thermo Scientific, Waltham, MA) and the lysate was cleared by centrifugation (16,000 × rpm, for 60 min at 4˚C). GST fusion proteins in the supernatant were filter-sterilized (0.22 μm) and purified using an immobilized glutathione Sepharose column (Cytiva, Marlborough, MA) under native conditions according to the manufacturer's instructions. Antigens were eluted by the high concentration of reduced glutathione (GSH) (Sigma-Aldrich, St. Louis, MO), and then proteins were concentrated using 10 kDa Centrifugal Filter Units (Millipore Sigma, Burlington, MA). All elutions were subjected to SDS-PAGE, and concentration was determined using the Nanodrop (Thermo Scientific, Waltham, MA) and Bradford assay.

## Mass spectrometry analysis

The purified protein lysate was resolved on NuPAGE 10% Bis-Tris Gel (Life Technologies, Carlsbad, CA), target band (~40 kDa and ~70 kDa size) was excised and processed for in-gel digestion using trypsin enzyme. The tryptic peptides were analyzed on nano-LC 1000 system (Thermo Fisher Scientific, San Jose, CA) coupled to Orbitrap Fusion mass spectrometer (Thermo Fisher Scientific, San Jose, CA). The peptides were loaded on a two-column setup using a pre-column trap of 2 cm x 100 μm size (Reprosil-Pur Basic C18 1.9 μm, Dr. Maisch

GmbH, Germany) and a 20 cm x 75 μm analytical column (Reprosil-Pur Basic C18 1.9 μm, Dr. Maisch GmbH, Germany) with a 110 min gradient of 2–30% acetonitrile/0.1% formic acid at a flow rate of 200 nl/min. The eluted peptides were directly electro-sprayed into mass spectrometer operated in the data-dependent acquisition (DDA) with top 35 mode. The full MS scan was acquired in Orbitrap in the range of 300–1400 m/z at 120,000 resolutions followed by MS2 in Ion Trap (HCD 30% collision energy) with 5 sec dynamic exclusion time. The RAW file from mass spectrometer was processed with Proteome Discoverer 1.4 (Thermo Scientific) using Mascot 2.4 algorithm (Matrix Science) with Fixed Value PSM validator against the recombinant GST-SinH protein sequence. The precursor ion tolerance and product ion tolerance were set to 20 ppm and 0.5 Da respectively. Maximum cleavage of 2 with Trypsin enzyme, dynamic modification of Oxidation on methionine, protein N-terminal Acetylation and Destreak on cysteine was allowed.

## Prediction of protein structure for SinH with AlphaFold2

The nucleotide sequence of SinH was used to recreate the translated amino acid sequence using ExPASy. All six possible reading frames (three forward, three backward) were generated and the frame that had the sequence for complete SinH was used as the amino acid sequence for structure prediction. ColabFold's AlphaFold2-Advanced Google Notebook (Google, Mountain View, CA) was used to generate predictions from amino acid sequence [44]. For multiple sequence alignment (MSA) necessary to build the consensus model for the structure of SinH, we used DeepMind's (DeepMind, London) original MSA jackhammer database previously generated for CASP14 using the complete Protein Data Bank (PDB) structure library [44–46]. Five prediction runs were run, with each run using a randomly chosen initiation point for the start of prediction runs. These models were ranked using the following two metrics: 1. pLDDT (predicted lDDT-Cα) with its ability to quantify the confidence of model per residue calculated by utilizing distances between Cα atoms in multiple reference models, and 2. AlphaFold-generated PAE (Predicted Aligned Error) for every residue, a numerical value of expected position error per residue [47]. The model with highest average pLDDT and lowest PAE was chosen as the best predicted structure of SinH. This structure was compared against previously solved structures of proteins deposited on PDB with similar functions by aligning spatial coordinates of models through RCSB Structural Alignment webserver, with the jFAT-CAT-rigid algorithm for alignment and TM-score as the metric for assessing alignment quality [48–50]. UCSF ChimeraX was used for analyzing structural features of the predicted model, determining local physical properties within domains, and visualizing the model [51]. BioRender was used for annotating models. Structure Prediction: SinH full length [dataset]. 2023 Jan 6 [cited 2023 Jan 22]. Dryad Digital Repository. Available from https://datadryad.org/resource/doi:10.5601/dryad.jm63xsjfx [52].

## Sequence alignment

A total of 334 *sinH* nucleotide sequences were extracted using megaBLAST to align the ST131 reference *sinH* sequence with our previously published phylogroup database of 1,348 *E. coli* chromosomes [39,53]. Once the *sinH* sequences were extracted, they were translated and sequences with premature stop codons were removed, leaving 308 sequences. In addition to these strains, the *sinH* nucleotide sequence from 26 *sinH* positive *E. coli* strains available in the Maresso lab were also extracted and translated, and duplications between the two datasets were removed. As an outgroup, Salmonella SinH amino acid sequence was used (accession: WP_023204198.1). Extracted SinH amino acid sequences were then aligned using MAFFT (version 7.450) with default settings and the "auto" setting for algorithm selection. The

resulting amino acid alignment was then used to create a phylogenetic tree with RAxML (version 8) with the GAMMA BLOSUM62 protein model and the Rapid Bootstrapping algorithm with 100 replicates [54]. The resulting trees were then used to create a consensus tree with 50% support threshold using the Consensus Tree Builder software in Geneious version 2022.0 created by Biomatters. The consensus tree was then annotated in BioRender. The SinH amino acid sequence from strains available to the lab were also aligned and a phylogenetic tree created using the same MAFFT and RAxML method outlined above. This alignment was exported from Geneious and annotated using BioRender.

## Experimental animals

The mouse strain used in this study was BALB/cJ mice (Jackson Laboratories, Bar Harbor, ME). All mice were female, 6 weeks of age. They received sterile food and water ad libitum and were housed 3–4 in filtered cages. All methods performed on mice were approved in accordance with relevant guidelines and regulations from "The Guide and Care and Use of Laboratory Animals" (National Institute of Health) and approved by Baylor College of Medicine's Institutional Animal Care and Use Committee (AN-5177).

## Vaccination

Purified proteins were mixed with alum adjuvant (G-Bioscience, St. Louis, MO) at a ratio of 2:1 (Antigen/adjuvant) according to the manufacturer's recommendations. Six-week-old female BALB/cJ mice were given three subcutaneous injections of 50 μg antigens on days 0, 14, and 28. Control groups were vaccinated with equivalent doses of GST (50 μg), alum adjuvant (30 μl), LPS (lipopolysaccharides, 3 EU, Thermo Scientific, Waltham, MA), or unvaccinated [55].

## Murine model of ST131 bacteremia

ExPEC sequence type 131 (ST131) strains, JJ1886, JJ2050, and JJ2547, were grown under the indicated conditions the day before injection. On the day of injection (day 42), the optical density (OD) was measured using a spectrophotometer set to 600 nm, and the overnight ExPEC strains were subcultured in LB broth at the ratio of 1:100 to an $OD600 \approx 0.6$ (Log phase, $\sim 1 \times 10^8$ CFU/ml). Then ExPEC strains were harvested by centrifugation ($3,500 \times g$ for 20 min at 4°C) and resuspended in equivalent 1× PBS. Mice were injected intraperitoneally by 50 μl of one of the ExPEC strains suspension ($5 \times 10^7$ CFU) on day 42 [56]. The inoculum was quantified by plating dilutions onto LB agar. After twenty-four hours, mice were euthanized and necropsied to collect their kidney, spleen, and liver. Organs were homogenized in 1 ml 1× PBS using BeadBlaster Refrigerated Homogenizer (Benchmark Scientific Inc, Sayreville, NJ, USA) and organ homogenates were plated on LB agar plates and incubated at 37°C to determine the number of bacteria or CFU per milliliter (mL). The schematic diagram was made in BioRender.

## Murine model of ST131 mortality study

For the mortality study, ExPEC sequence type 131 (ST131) strain JJ2050 was grown under the indicated conditions the day before injection as described above. On the day of injection (day 42), mice were injected intraperitoneally with 50 μl of the ExPEC strain JJ2050 suspension ($5 \times 10^7$ CFU) [56]. Mice were monitored twice a day for 10 days. Murine survival was followed with time, and moribund animals were euthanized/necropsied to determine bacterial levels in the kidneys, spleen, and liver. The organs were homogenized, and the JJ2050 bacterial

load in the infected organs was quantified by the determination of CFU. The schematic diagram was made in BioRender.

## Murine model of acute urinary tract infection (acute UTI)

UPEC strains, UTI89 and CFT073, were grown and prepared under the indicated conditions. On day 42, Mice were inoculated transurethrally by 50 μl of one of the UPEC strains suspension ($10^8$ CFU) as described previously [57]. The inoculum was quantified by plating dilutions onto LB agar. After twenty-four hours, mice were euthanized and necropsied to collect bladders. Bladders were homogenized in 500 μl 1× PBS using BeadBlaster Refrigerated Homogenizer and organ homogenates were plated on LB agar plates and incubated at 37˚C to determine the number of bacteria or CFU per milliliter (mL). The schematic diagram was made in BioRender.

## Murine model of GI tract colonization in healthy mice

ExPEC sequence type 131 (ST131) strains were grown and prepared under the indicated conditions. Mice were subjected to gavage with 100 μl of a bacterial suspension ($10^9$ CFU) with a sterile (20-gauge, 38-mm-long) flexible needle on day 42. The inoculum was quantified by plating dilutions onto LB agar. After twenty-four hours, mice feces were collected and homogenized in 1 ml 1× PBS using BeadBlaster Refrigerated Homogenizer (Benchmark Scientific Inc, Sayreville, NJ, USA) and feces homogenates were plated on LB agar plates and incubated at 37˚C to determine the number of bacteria or CFU per milliliter (mL). The schematic diagram was made in BioRender.

## Murine model of GI tract colonization in immunosuppressed mice

ExPEC sequence type 131 (ST131) strains were grown and prepared under the indicated conditions. Mice were subjected to gavage with 100 μl of a bacterial suspension ($10^9$ CFU) with a sterile (20-gauge, 38-mm-long) flexible needle on day 42. The inoculum was quantified by plating dilutions onto LB agar. Then cyclophosphamide (Cytoxan [CTX]) (United States Pharmacopeia) was dissolved in sterile water and diluted with filter-sterilized 1× PBS to a final concentration of 10 mg/ml, and the mice were given a total dose of 450 mg/kg of body weight (three 150-mg/kg doses administered at 1-day intervals (day 43, 45, 47) intraperitoneally (i.p.) at the indicated time points [56,58]. On the day of 48, mice feces were collected and homogenized in 1 ml 1× PBS using BeadBlaster Refrigerated Homogenizer and feces homogenates were plated on LB agar plates and incubated at 37˚C to determine the number of bacteria or CFU per milliliter (mL). The schematic diagram was made in BioRender.

## Murine model of multiple sequence-type (ST) model

Different sequence-type (ST) of ExPEC strains were grown and prepared under the indicated conditions as described in the previous model. Mice were injected intraperitoneally with 50 μl of a different sequence-type (ST) *E. coli* suspension, either ST73-mixture (Mix of CFT073, W0040, W0088, W0116 equally) or ST95-mixture (Mix of UTI89 and W0060 equally) (in total $5 \times 10^7$ CFU of each mixture), on day 42 [56]. The inoculum was quantified by plating dilutions onto LB agar. Mice were monitored twice a day for 5 days, and moribund animals were euthanized/necropsied to determine bacterial levels in the kidneys, spleen, and liver. Organs were homogenized in 1 ml 1× PBS using BeadBlaster Refrigerated Homogenizer and organ homogenates were plated on LB agar plates and incubated at 37˚C to determine the number of bacteria or CFU per milliliter (mL). Moribundity was determined through the observation of

multiple features, including rough coat, hunched posture, lethargy and hyperpnea. The schematic diagram was made in BioRender.

## ELISA

For the indirect enzyme-linked immunosorbent assay (ELISA), 100 μl of 20 μg/ml or 2 μg/ml purified proteins were coated onto Thermo Fisher 96-well Nunc plates and incubated at 4˚C overnight. The plate was washed three times by flooding all wells with wash buffer (0.05% Tween 20 in 1× PBS), and nonspecific binding sites were blocked with 150 μl 5% milk solution in 1× PBS for 2 hours. Serum was taken from individual mice after complete immunization and ExPEC infection and urine were taken from individual mice after complete immunization. Then the wells were coated with serum diluted 1:5000 in 5% milk or 50 μl undiluted urine, and the plate was incubated with gentle rocking overnight at 4˚C. The following day, the plate was washed three times with wash buffer, and then a volume of 100 μl secondary antibodies (anti-mouse IgG generated in rabbit conjugated to horseradish peroxidase, diluted in 1:5000 in 1× PBS or anti-Mouse IgA Cross-Adsorbed Secondary Antibody generated in goat, diluted in 1:2000 in 1× PBS) was added into each well, and entire sample gently rocked at 4˚C for 1 hr. The plate was washed 3 times with wash buffer and 1× PBS before 100 μl TMB (3,3',5,5'-Tetramethylbenzidine) solution was added to the wells and allowed then incubate at room temperature for 5 to 10 minutes until color developed. The reaction was stopped by adding 50 μl 2M sulfuric acid ($H_2SO_4$) to the well. The absorbance of each well was measured at 450 nm by using the BioTek Synergy HT plate reader [55]. All experiments were performed with three replicates, and ELISA readouts were normalized to the same mole.

## Statistical analyses

Graphing and statistical analyses were performed using Graphpad Prism version 9 (GraphPad Software, Inc.). Significance was determined using the Kruskal-Wallis analysis of variance (ANOVA) with Dunn's multiple comparisons correction. All survival curves were compared using the Genhan-Breslow-Wilcoxon curve comparison. All statistics were conducted using 95% confidence intervals, alpha values were set to 0.05 and statistical significance was determined if calculated P values were below 0.05. The lines of all the bar graphs were at the median with a 95% confidence interval (CI). One star (*) $P < 0.05$, two stars (**) $P < 0.01$, three stars (***) $P < 0.001$, four stars (****) $P < 0.0001$. The Box-and-whisker plots and Kaplan Meier survival curves were exported from Graphpad Prism 9 and annotated using BioRender.

## Dryad DOI

10.5601/dryad.jm63xsjfx [52]

## Results

### Identification of SinH as vaccine target

Pathogenic *Escherichia coli* is a significant cause of global human morbidity and mortality. The overarching vaccine challenge with this pathogen is its propensity to readily take in or lose genes associated with antibiotic resistance and virulence, in addition to a pangenome that deviates by as much as 30% between strains [59,60]. Using a comparative genomics approach, we previously reported an analysis of *sinH* prevalence amongst *E. coli* pathotypes, phylogroups, and sequence types (**Fig 1A**). Pathotypes are groups of pathogenic strains that share the same phenotype of the disease, which broadly can divide into extraintestinal pathogenic *E. coli* (ExPECs) or intestinal pathogenic *E. coli* (InPECs) [4–6]. *E. coli* also is characterized by their

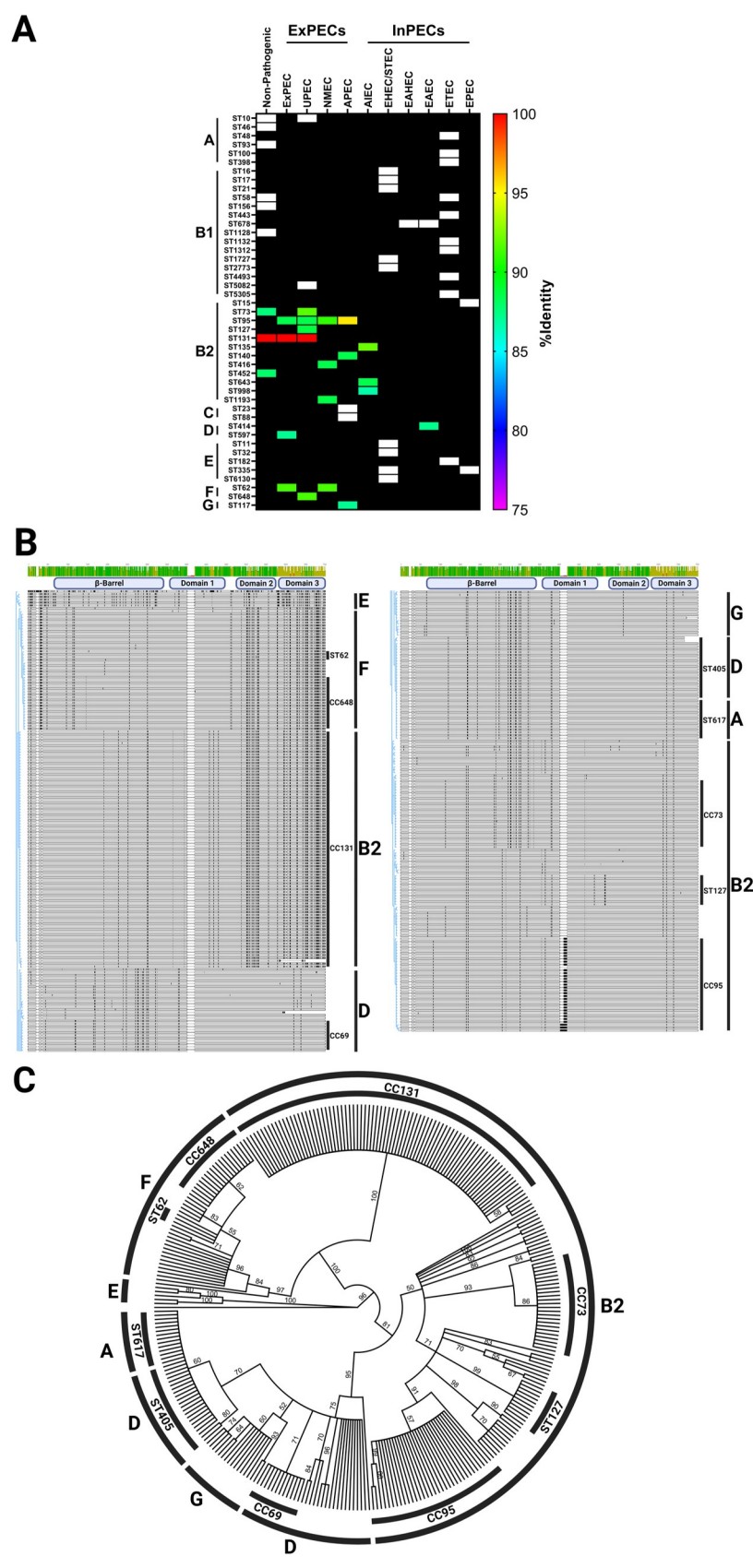

**Fig 1. Comparative genomics heatmap, amino acid sequence alignment, and phylogenetic tree of *sinH* sequence.**
(A) Pathotype, phylogroup, sequence types of distribution of *sinH* sequence. Heatmap showing nonpathogenic *E. coli*, ExPECs and InPECs. Columns are organized by pathotypes, and rows are organized first by phylogroups, then by sequence types. Each cell in the heatmap is colored based on percent nucleotide identity compared to the reference used to generate the alignments, and the black boxes indicate there is no sequence type (ST) present for the listed pathotype whereas white boxes indicate there is a sequence type but it does not contain a sinH homolog. (B) MAFFT alignment of the amino acid sequence of SinH. Alignment is annotated with phylogroup and sequence type. An identity histogram is shown at the top, and black represents amino acid differences from the majority consensus. (C) Consensus maximum-likelihood phylogenetic tree of SinH generated from alignment shown in Fig 1B using RAxML and rooted with Salmonella SinH. Branch labels indicate percentage support from 100 rapid bootstrap replicates. The consensus tree and alignment were annotated in BioRender.

genetic lineage into phylogenetic groups, of which there are four major (A, B1, B2, and D) and five minor (C, E, F, G, and cryptic clade I) [61–63]. The sequence types (STs) were established following the multilocus sequence typing (MLST) scheme of Achtman [64–66]. SinH was strongly associated with ExPECs pathotypes and also has a tight association with phylogroup B2/D/F/G (**Fig 1A**). B2 and D strains from this clade are a major source of ExPEC infections [67], suggesting that SinH might be associated with ExPEC infections directly or indirectly. In addition, the *sinH* sequence is also detected in B2 commensal strains, but not detected in A and B1 phylogroup commensal strains, which are the source of most human commensal *E. coli* strains [68]. Furthermore, our previous work verified the association of the *sinH* sequence with the phylogroup B2, where it is found in 98% of strains, but is also strongly associated with other phylogroups, such as phylogroup F (100%), G (100%), and D (67%) [39]. The B2 *E. coli* strains harboring the *sinH* sequence included all members of sequence type 131 (ST131), sequence type 73 (ST73), sequence type 95 (ST95), and sequence type 127 (ST127). These sequence types have become the most prevalent and common lineages in *E. coli* isolates that were recovered from the hospital and community patients with bacteremia or UTI worldwide [69–73]. Furthermore, we observed a phylogenetic relationship between all SinH protein sequences found in the previously curated database of 1,348 complete *E. coli* chromosome sequences [39]. SinH encoded by ST131 *E. coli* appear to have somewhat diverged compared to other ExPEC causing-sequence types of the B2 phylogroup such as ST73 (88.6% identical, 93.2% similar), ST95 (88.8% identical, 92.9% similar), and ST127 (90.6% identical, 95.2% similar), especially in domain 3 (**Fig 1B and 1C**). The ST131 allele appears to be nearly identical to those found in the F phylogroup, including the ExPEC-causing ST62 and ST648 sequence types, whereas SinH from other B2 strains is more similar to those from phylogroup D and G. The determination of all pathotypes, phylogroups and sequence type (ST) listed above was carried out by the scheme of our lab previous work [39]. The *sinH* sequences were located using megaBLAST (version 2.11). BLAST hits were elongated to the first stop codon, extracted, translated, and then aligned using MAFFT (version 7.450) (**Fig 1B**). The resulting alignment was used to create a phylogenetic tree was created using Geneious Prime's Consensus Tree Maker using RAxML(version 8.2.11) trees with 100 bootstrap replicates (**Fig 1C**). Together, these results indicate that, while *sinH* has been evolving across different phylogroups, there is conservation within the same phylogroup.

## Structural and functional analysis of SinH

Recent work suggests that SinH shares a similar structural and evolutionary history with intimin and invasin as a virulence-associated bacterial outer membrane protein [74,75]. We used AlphaFold2 to predict the structure of full-length SinH and compared the structure (blue) to solved structures of *Y. pseudotuberculosis* invasin and Enterohemorrhagic (EHEC) *Escherichia coli* intimin to gain insights into the function of SinH in host-pathogen interactions. To

determine the characteristics of surface-exposed of SinH, we used as Pairwise Structure Alignment at RCSB with the jFATCAT-rigid algorithm to align predicted SinH structures against existing structures from the Protein Data Bank (PDB) [48,49]. The predicted structure of full-length SinH is organized into four distinct domains (from left to right): translocation β-barrel transmembrane domain (purple), Ig-like domain-1 (green), Ig-like domain-2 (red), and Ig-like domain-3 (referred as the Receptor binding domain hereafter, blue). Also shown here is the calculated electrostatic density map (blue: positive charge, red: negative charge) for SinH (**Fig 2A**). To quantify the alignment of each SinH domain to known structures, we used a template modeling score (TM-score), a metric for assessing topological similarity of protein folds as calculated by distances between corresponding amino acid residues, which ranges in value from 0 to 1 with scores greater than 0.5 indicating two proteins generally having the same fold [76]. The TM-score for transmembrane β-barrel domains of SinH and *Y. pseudotuberculosis* invasin was 0.96 (PDB: 4E1S), while SinH and EHEC intimin was 0.95 (PDB: 4E1T) [77]. The SinH transmembrane domain also had amino acid sequence similarities of 66% to invasin and 62% to intimin, and these numbers are reflected upon the same fold these proteins assume (**Fig 2B**). The closest match to Ig-like domain 1 of SinH was domain-3 of *Y. pseudotuberculosis* invasin (PDB: 1CWV), with a TM-score of 0.54 that indicated high likelihood of the same protein fold [78]. Unlike transmembrane domains, the amino acid makeup of these structurally related domains differed considerably, with low sequence identity of 9% and similarity of 29%, which suggests divergence in amino acids between these two proteins that nonetheless conserved the structure fold (**Fig 2C**). This disparity between the fold and amino acid composition preservation was also observed when comparing Ig-like domain-2 of SinH to the domain-3 of *Y. pseudotuberculosis* invasin, with a TM-score of 0.55 with amino acid sequence identity of 13% and similarity of 24% (**Fig 2D**). This suggests that domain-2 also follows the trend of domain-1 in structural similarity and sequence dissimilarity to invasin. Given that the structures of domains 1 and 2 both matched to invasin domain 3, we aligned domains 1 and 2 of SinH against each other to confirm that these two domains have similar folds (DOI: https://doi.org/10.5061/dryad.jm63xsjfx). These two domains indeed had very similar structures (TM-score of 0.5), hinting that the role of domains 1 and 2 in SinH is to serve as a scaffold for positioning the receptor binding domain (RBD). In the Ig-like domain-3 (RBD) of SinH, a lectin-like domain is observed which incidentally is not found in both invasin and intimin. We thus decided to run the alignment between only RBD of SinH against its counterpart domains in invasin and intimin (PDB: 1F00) with a lectin-like domain to determine similarities in these analogous domains [79] (**Fig 2E**). Matching RBD to invasin gave TM-score of 0.45 with sequence identity of 10% and sequence similarity of 24%, while intimin gave TM-score of 0.41 with sequence identity of 8% and sequence similarity of 25%. On the other hand, aligning these domains from solved structures of invasin and intimin yielded high TM-score of 0.71 with sequence identity of 21% and sequence similarity of 37% (DOI: https://doi.org/10.5061/dryad.jm63xsjfx). These results suggest that the SinH RBD is phylogenetically more distant from invasin and intimin than these two are to each other as far as structure is concerned, a finding that hints that RBD of SinH may have a different function and target than invasin and intimin, where this Ig-like domain may participate in the binding of SinH to a novel receptor on the host. In summary, our alignment statistics of the predicted SinH structure indicate high structural similarity of the SinH to intimin and invasin even with very poor sequence homology, thus explaining why there are few reports linking these two genes expressing structurally very similar proteins. Our predictions of the surface-exposed Ig-like domains of SinH were supported by a recently published work, which demonstrated IatB (SinH) is located on the cell surface and contributes to biofilm formation [80].

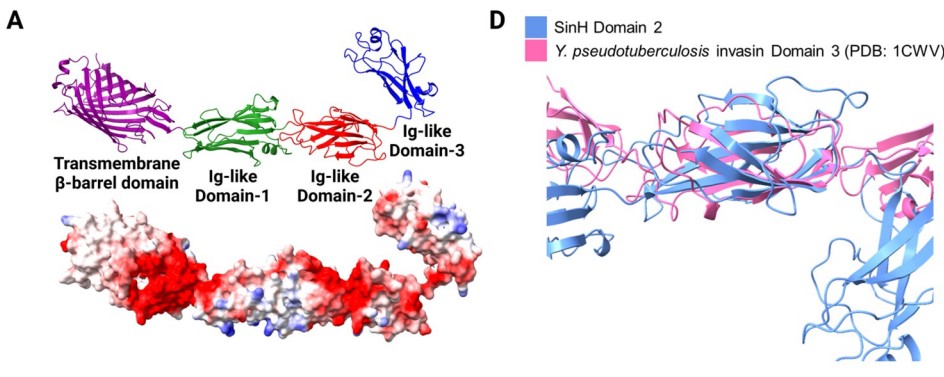

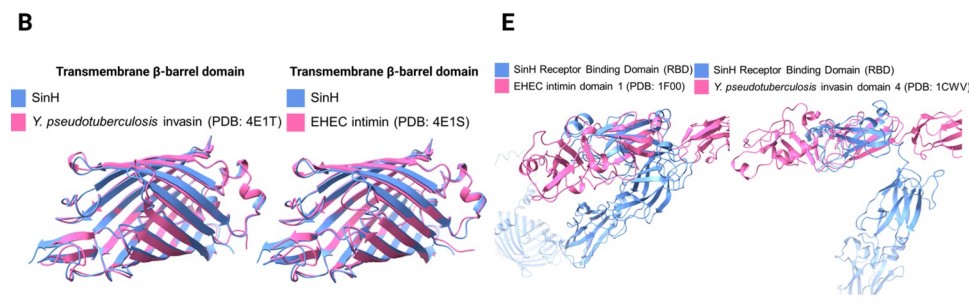

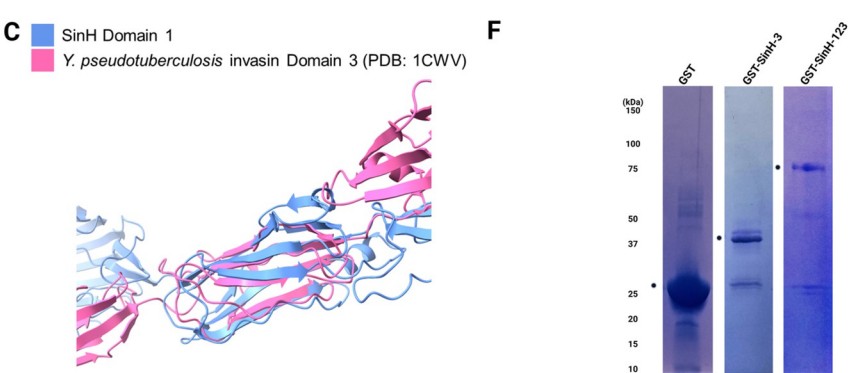

**Fig 2. Structural alignment of predicted full-length SinH and expression and purification of SinH-based candidate antigens.** Structural alignments were generated by Pairwise Structure Alignment webserver, and aligned structures were visualized using ChimeraX and annotated with BioRender. (A) Predicted structure of full-length SinH protein (excluding disordered residues 1 through 101) with four distinct domains (Translocation β-barrel transmembrane domain: purple, Ig-like domain-1: green, Ig-like domain-2: red, Ig-like domain-3 (Receptor binding domain): blue). (B) Alignment between transmembrane β-barrel domains of predicted SinH protein structure (blue) and transmembrane domains of *Y. pseudotuberculosis* invasin (PDB: 4E1T) (red, left) and EHEC intimin (PDB: 4E1S) (red, right). (C) Alignment between domain-1 of SinH (blue) and domain-3 of *Y. pseudotuberculosis* invasin (red). (D)

Alignment between domain-2 of SinH (blue) and domain-3 of *Y. pseudotuberculosis* invasin (red). (E) Alignment between the receptor-binding domain (RBD) of SinH (blue) and Ig-like domain-1 of EHEC intimin (red, left) and Ig-like domain-4 of *Y. pseudotuberculosis* invasin (red, right). (F) Genes encoding SinH-based antigens (Ig-like domain-1,2,3 or Ig-like domain-3) were cloned from ExPEC ST131 strain JJ1887. SinH-based antigens were recombinantly expressed with a glutathione-S-transferase (GST) tag and purified using immobilized GST-affinity chromatography. Purified antigens were separated by SDS-PAGE and stained with Coomassie blue stain buffer. Predicted sizes of tagged proteins are as follows: GST-SinH-3, 40 kDa; GST-SinH-123, 70 kDa. Circle symbols indicate the locations of the GST-SinH Domain-3 and GST-SinH Domain-123, respectively, for each individual gel. The SDS-PAGE were annotated in BioRender.

## SinH-based candidate antigens expression and purification

In preparation for immunization, the genes for SinH-based antigens, SinH-3 (Ig-like domain-3 fragment of SinH, amino acid residues 602 to 724) and SinH-123 (Ig-like domain-1,2 and 3 fragments of SinH, amino acid residues 337 to 724) were cloned as glutathione-S-transferase (GST)-tagged fusions, expressed, and purified under native conditions as N-terminal GST-tagged recombinant proteins. Bacterial cultures expressing recombinant vaccine antigens were lysed, and supernatant which contained the SinH-antigens were collected. Recombinant proteins were purified by GST-affinity chromatography and visualized by SDS-PAGE. Two dominant bands assumed to be GST-SinH-3 and GST-SinH-123 (40 kDa, 70 kDa) were observed after Coomassie blue staining of the gels (Fig 2F). To verify their authenticity, the putative SinH-3 and SinH-123 proteins were subjected to Mass Spectrometry (S1 Fig). Purified protein bands were resolved and digested in gel. The tryptic peptides were analyzed on nanospray LC-MS (liquid chromatography-mass spectrometry) system. The eluted peptides were directly electro-sprayed into mass spectrometer and analyzed by data-dependent acquisition (DDA). For the GST-SinH-3, the coverage (the percentage of the protein sequence by identified peptides) was approximate 98%; and for the GST-SinH-123, the coverage was approximate 97% (S1 Fig). In summary, high sequence coverage was detected in each band and was sufficient to confirm the identity of both GST-SinH-3 and GST-SinH-123 recombinant proteins (GST-SinH-3, in short SinH-3 in following; GST-SinH-123, in short SinH-123 in following).

## Immunization with SinH-based antigens confers protection against ExPEC sequence type 131 (ST131) bacteremia

ExPEC ST131 strains are drug-resistant and are responsible for millions of global antimicrobial-resistant (AMR) infections annually and comprise a significant risk of bloodstream infections worldwide [15,16]. To verify the protective efficacy of SinH-based antigens in a systemic model of ExPEC ST131 bacteremia, purified antigens were mixed with alum as an adjuvant at a ratio of 2:1 (antigen/alum), followed by subcutaneous immunization of mice with either antigens (SinH-3 or SinH-123) or GST alone on days 0, 14, 28. Mice were then intraperitoneally injected on day 42 with three ExPEC ST131 strains, JJ1886, JJ2050, or JJ2547 ($5 \times 10^7$ CFU). These strains were chosen because they represent a diverse range of strains from the genetically distinct and epidemic clade—clade C2, or H30Rx—of the ST131 clonal group [81]. The vaccination schematic used in this experiment is shown in Fig 3A. The infection was allowed to progress for 24 hours before the mice were euthanized and their kidney, spleen, and liver collected. The organs were homogenized, and the ExPEC bacterial load in the infected organs was quantified by the determination of CFU (Fig 3B–3D). Combining the counts from all cohorts (as a way to assess the total effect of vaccination across all organs and strains), both SinH-based vaccines showed a clear and statistically significant reduction in bacterial burden (Adjusted *P* value, SinH-3, $P<0.0001$; SinH-123, $P<0.0001$). In comparison to the mice that received GST alone, SinH-3 vaccinated mice had a 55-fold decrease, and SinH-123 vaccinated

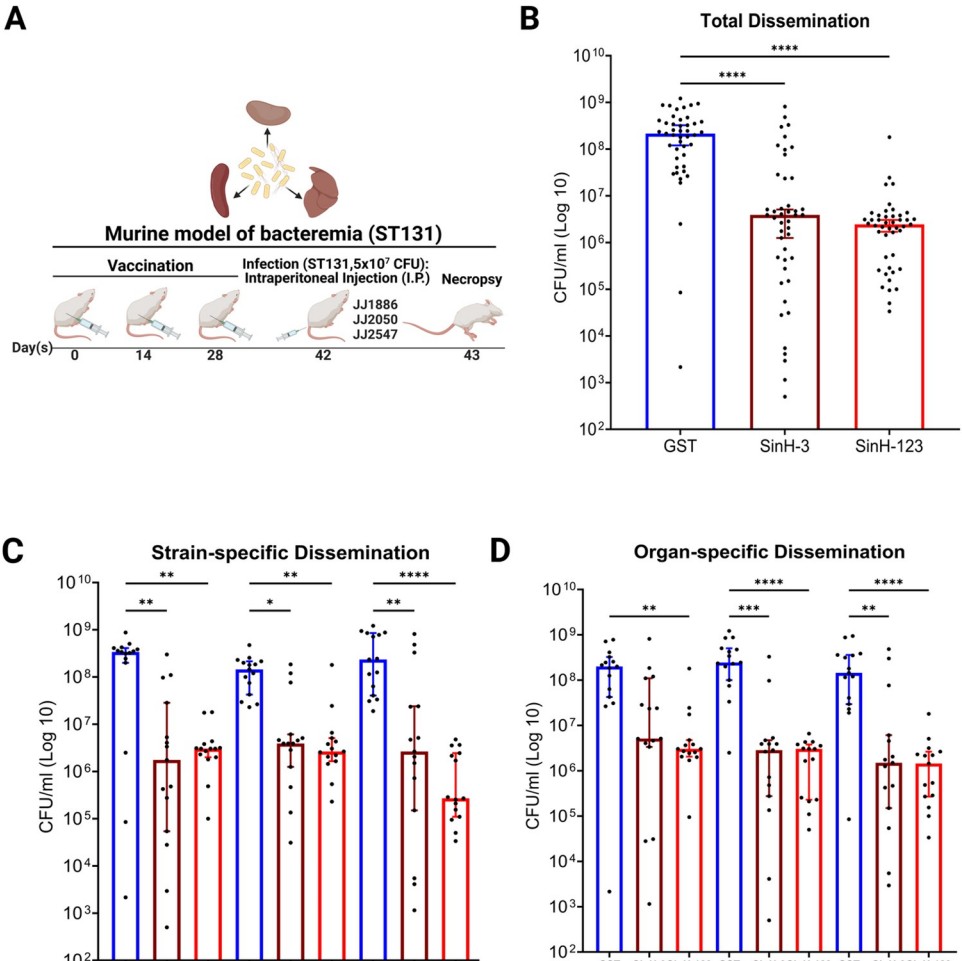

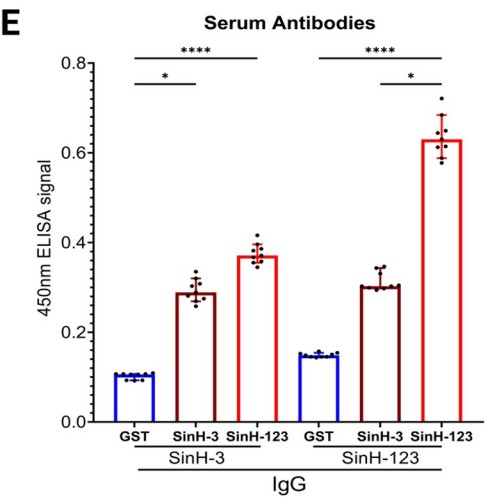

**Fig 3. Assessment of the protective efficacy and immunogenicity of SinH-based vaccines against ExPEC sequence type 131 (ST131) bacteremia.** (A) The vaccination scheme was used in this experiment. BALB/cJ, 6 weeks old, female mice were subcutaneously immunized with SinH-based antigens (SinH-3, SinH-123, N = 15) or GST alone (N = 15)

and injected with an intraperitoneal (IP) injection of $5 \times 10^7$ CFU of different ExPEC ST131 strains (JJ1886, JJ2547, JJ2050). Organs were harvested and plated to determine bacteria levels. Serum was taken from individual mice after immunization and ExPEC infection. The schematic diagram was made in BioRender. (B) Box-and-whisker plots of the bacterial levels (CFU/ml) in combining the counts from all organs (liver, spleen, kidney) and all ExPEC strains (JJ1886, JJ2547, JJ2050); (C) or the bacterial levels (CFU/ml) of each ExPEC ST131 strain in combining the counts from all organs; (D) or the bacterial levels (CFU/ml) of all ExPEC strains in each type of organ following necropsy. (E) ELISA analysis of sera from SinH-based antigens vaccinated animals using antigens, SinH-3 or SinH-123 (GST-tag removed), as the capture antigen. Error bars indicate the median with 95% confidence interval (CI). Significant was determined by theKruskal-Wallis analysis of variance (ANOVA) with Dunn's multiple comparisons correction. Symbols represent data of individual mice. One star (*) $P < 0.05$, two stars (**) $P < 0.01$, three stars (***) $P < 0.001$, four stars (****) $P < 0.0001$. The Box-and-whisker plots were exported from Graphpad Prism 9 and annotated using BioRender.

mice had an 88-fold decrease in the median levels of total ExPEC (**Fig 3B**), thereby demonstrating the results were rigorous across multiple strains, multiple organs, and for at least two antigens of the target autotransporter. A reduction in bacterial levels was also observed in all three organs when each bacterial strain was parsed into separate data (**Fig 3C**) thereby suggesting the reduction was likely regardless of the genetic background of this sequence type. For example, these reductions were for SinH-3 37-fold (194.29, 37.18, and 88.68) and for SinH-123 at least a 54-fold reduction (111.48, 54.72, and 870.37) in the median level of each ExPEC strain. In addition, in combining the counts from all ExPEC strains, mice vaccinated with SinH-based antigens significantly reduced bacterial levels in each type of organ. For example, SinH-3 vaccinated mice had at least a 38-fold reduction (38.83, 85.96, and 96.67), and SinH-123 vaccinated mice had at least a 65-fold reduction (65.57, 80.33, and 100) in the median level of ExPEC for the liver, spleen and kidney compared to the GST-only control (**Fig 3D**), a result that was equivalent for both SinH antigens.

In addition, to determine whether subcutaneous immunization with SinH-based vaccines induces a vaccine-specific humoral immune response, serum samples were collected from each mouse on the day of euthanasia. The levels of vaccine antigen-specific serum IgG were quantified via indirect ELISA. Of note, SinH-123 induced statistically significant production of IgG specific for the immunogen over the GST control (**Fig 3E**, Adjusted $P$ value, $P<0.0001$, $P<0.0001$). When SinH-3 (GST-tag removed) was used as the ELISA antigen, SinH-3 vaccinated mice showed higher antigen-specific serum IgG responses than the control group mice (Adjusted $P$ value, $P = 0.0319$) (**Fig 3E**). Interestingly, vaccination with SinH-123 produced the most robust serum IgG response, perhaps due to the construct being composed of three domains, thereby providing more antibody-recognition sites than SinH-3. This result might explain the reason SinH-123 antigens demonstrated more stable protection against the colonization of ExPEC sequence type 131 in the murine model of bacteremia, which positively correlates to the protective efficacy of antigen.

## Immunization with SinH-based antigens reduces the mortality of ExPEC ST131 bacteremia

Next, we determined whether subcutaneous immunization with SinH decreased the mortality of the vaccinated mouse after being challenged by the ST131 *E. coli*. We used alum-only (30 μl/mouse) and LPS-only (3 EU/mouse) as the control group. LPS-only or alum-only was added in this experiment to control for the possibility that endotoxin or adjuvant might contribute to the overall protection observed in SinH vaccinated animals. Mice were vaccinated with SinH antigens (SinH-3 or SinH-123), alum, or LPS, followed by intraperitoneal injection on day 42 with ST131 JJ2050 *E. coli* strain ($5 \times 10^7$ CFU). Mice were monitored twice a day for 10 days. Murine survival was followed with time, and moribund animals were euthanized/necropsied to determine bacterial levels in the kidneys, spleen, and liver. The organs were homogenized,

and the JJ2050 bacterial load in the infected organs was quantified by the determination of CFU. The vaccination schematic used in this experiment is shown in **Fig 4A**. The results showed that mice vaccinated with an LPS-only or alum-only control died within 1 d.p.i. In contrast, animals immunized with either of the SinH antigens demonstrated a survival rate of 33.3% after 10-days (Adjusted *P* value, SinH-3, *P* = 0.0037; SinH-123, *P* = 0.0090) (**Fig 4B**).

The bacterial levels in SinH vaccinated animals showed results consistent with the survival data. Combining the counts from all organs, compared to the mice vaccinated with LPS-only (moribund within 1 d.p.i), the mice vaccinated with SinH-3 significantly reduced the bacterial burden of JJ2050 in organs after 2 d.p.i (moribund within 2 d.p.i, Adjusted *P* value, *P* = 0.0248) and after 10 d.p.i (surviving mice, Adjusted *P* value, *P*<0.0001). Also, compared to the mice vaccinated with alum-only, the mice vaccinated with SinH-3 significantly reduced the bacterial burden of JJ2050 in organs after 10 d.p.i (surviving mice, Adjusted *P* value, *P*<0.0001). In comparison to the bacterial level of the mice that moribund within 1 d.p.i which received LPS-only or alum-only, those surviving SinH-3 vaccinated mice had a 4-log reduction or 3.8-log reduction in the median level of JJ2050 strain after 10 d.p.i (**Fig 4C**).

Likewise, compared to the mice vaccinated with LPS-only or alum-only (moribund within 1 d.p.i), the SinH-123 vaccinated mice also had significantly reduced bacterial burdens in organs after 2 d.p.i (moribund within 2 d.p.i, Adjusted *P* value, *P* = 0.0023, *P* = 0.0281) and 10 d.p.i (surviving mice, Adjusted *P* value, *P*<0.0001, *P*<0.0001). In comparison to the bacterial level of the mice that moribund within 1 d.p.i which received LPS-only or alum-only, those surviving SinH-123 vaccinated mice had a 6.6-log reduction or 6.4-log reduction in the median level of JJ2050 strain after 10 d.p.i (**Fig 4C**).

## Immunization with SinH-3 confers protection against the bacteremia of multiple ExPEC sequence types

Although ST131 is now a pandemic clonal lineage of ExPEC, other clonal ExPEC lineages, such as ST95 and ST73 were the second and third most common clonal ExPEC group isolated from urine and blood from patients with bloodstream infections [71–73]. MegaBLAST and MAFFT were used to align the *sinH* sequence from the ST95 and ST73 sequence types. A total of 30 amino acid mutations were observed in the domain-3 of the SinH sequence (**Fig 5A**) compared to ST131, the most varied domain of the three. To determine if SinH-3 is effective against multiple ExPEC sequence types in the murine model of bacteremia, mice were vaccinated with this domain as described in Fig 3, followed by intraperitoneal injection on day 42 with a mixture of strains of ST73 (CFT073) and ST73-like (W0040, W0088, W0116) equally or a mixture of strains of ST95 (UTI89) and ST95-like (W0060) equally (ST73-mixture or ST95-mixture, each measure total at $5 \times 10^7$ CFU—**Fig 5B**). On day 48, mice were euthanized, their liver, spleen, and kidney organs were collected and homogenized. The ExPEC bacterial load in the infected organs was quantified by the determination of CFU. Of the mice immunized with the SinH-3, at least 75% (Adjusted *P* value, ST73-mixture, *P* = 0.0221) and 86% (Adjusted *P* value, ST95-mixture, *P* = 0.0024) of the subjects survived the 5-day challenge period (**Fig 5C and 5D**), a number highly favorable compared to all the subjects failing to survive in the control cohort (0% survival). In addition, combining the counts from all organs, mice vaccinated with SinH-3 significantly reduced bacterial burden in organs of both ExPEC ST73-mixture and ST95-mixture (Adjusted *P* value, ST73, *P* = 0.0085; ST95, *P* = 0.0005) (**Fig 5E**). In comparison to the unvaccinated mice, SinH-3 vaccinated mice had an approximate 4-log reduction of ExPEC ST73-mixture strains and an approximate 4.3-log reduction of ExPEC ST95-mixture strains in the median level of ExPEC colonization (**Fig 5E**).

**A**

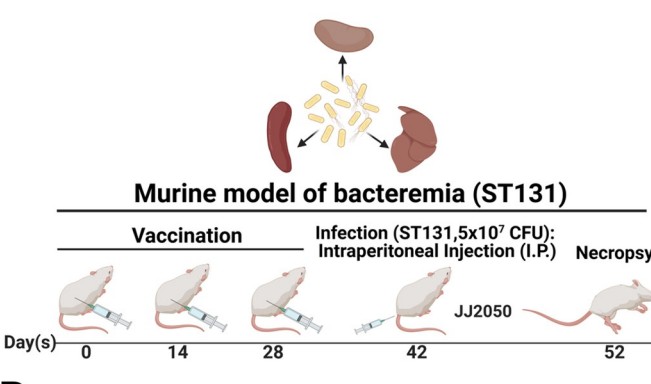

**B**

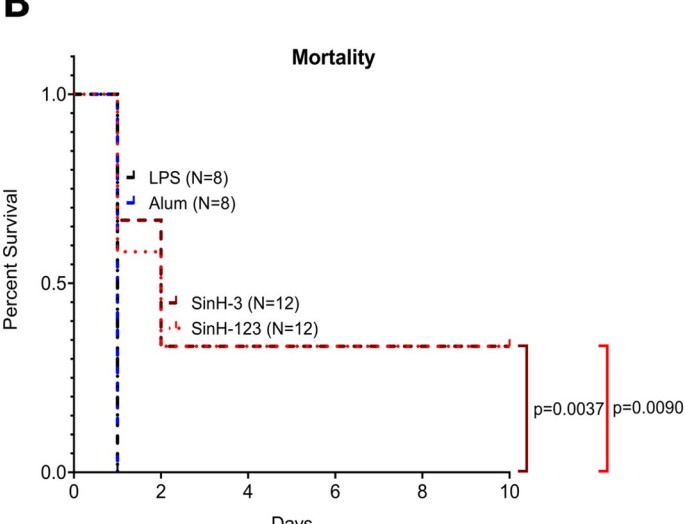

**C**

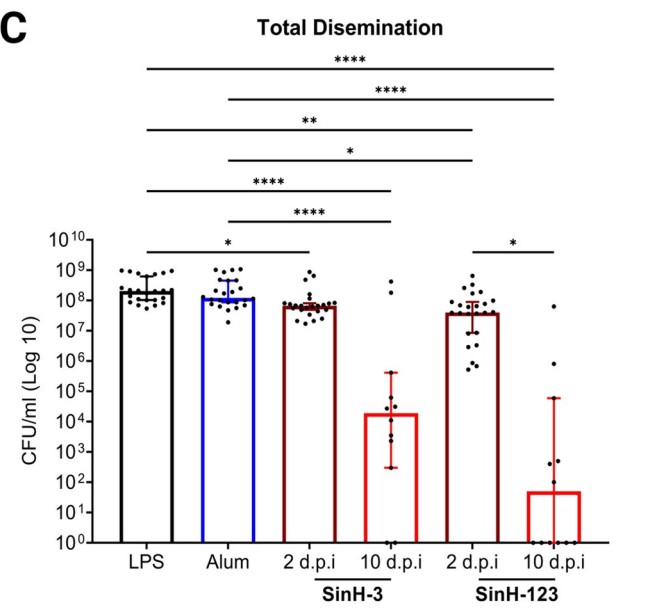

**Fig 4. Assessment of the protective efficacy of SinH-based vaccines reduced the mortality of ExPEC sequence type 131 (ST131) bacteremia.** (A) The vaccination scheme was used in this experiment. BALB/cJ, 6 weeks old, female mice were subcutaneously immunized with SinH-based antigens (SinH-3, SinH-123, N = 12), alum-only (N = 8) or LPS-only (N = 8) and injected with an intraperitoneal (IP) injection of $5 \times 10^7$ CFU of ExPEC ST131 strain JJ2050. Mice were monitored twice a day for 10 days, and moribund animals were euthanized/necropsied to determine bacterial levels in the kidneys, spleen, and liver. The schematic diagram was made in BioRender. (B) The survival rate of ST131 ExPEC strain JJ2050 was determined using the Gehan-Breslow-Wilcoxon comparison. (C) Box-and-whisker plots of the JJ2050 bacterial levels (CFU/ml) of the SinH-3 vaccinated group and SinH-123 vaccinated group in combining the counts from all organs (liver, spleen, kidney) at 2 d.p.i and 10 d.p.i. Error bars indicate the median with 95% confidence interval (CI). Significant was determined by the Kruskal-Wallis analysis of variance (ANOVA) with Dunn's multiple comparisons correction. Symbols represent data of individual mice. One star (*) $P < 0.05$, two stars (**) $P < 0.01$, three stars (***) $P < 0.001$, four stars (****) $P < 0.0001$. The Box-and-whisker plots and Kaplan Meier survival curves were exported from Graphpad Prism 9 and annotated using BioRender.

## Immunization with SinH-3 confers protection against cystitis in the murine model of acute UTI

Urinary tract infections (UTIs) are one of the most common diseases globally [82]. To examine the protective efficacy of SinH-based antigens against ExPEC in the murine model of acute UTI, female BALB/cJ mice were immunized subcutaneously on days 0, 14, and 28 as described in the previous model. Mice were transurethrally inoculated on day 42 with $10^8$ CFU of UPEC strains (UTI89 or CFT073, **Fig 6A**). After 24 hours, bladders were homogenized, and the UPEC bacterial load in the infected organs was quantified (**Fig 6B and 6C**). SinH-3 vaccination significantly protected the mice against UTI89 colonization. SinH-3 vaccinated mice had a 44-fold reduction in the median levels of UTI89 colonization in the bladder in comparison to the mice that were given GST alone (Adjusted *P* value, *P* = 0.0430), **Fig 6B**). Although an approximately 20-fold reduction in the median levels of UTI89 colonization in the bladder compared to the control group was observed for the three-domain antigen SinH-123, the effect was not statistically significant (Adjusted *P* value, *P* = 0.2843, **Fig 6B**). For the experimental UPEC strain CFT073, there was no difference between the experimental groups and the control group (**Fig 6C**). To evaluate the humoral immune response at the site of UPEC colonization, urine samples were collected from individual mice following a series of subcutaneous immunization with either SinH-based antigens or GST, and the levels of vaccine-specific urinary IgG and IgA were quantified via indirect ELISA. For the urinary IgG, SinH-3 vaccinated mice induced significantly higher levels of antigen-specific urinary IgG than those in the control group (Adjusted *P* value, *P*<0.0001; *P* = 0.0001). In addition, SinH-3 vaccinated mice demonstrated a higher level of urinary IgG than SinH-123 vaccinated mice, which might explain the reason that SinH-3 showed better protection against the colonization of UPEC in the murine model of acute UTI (**Fig 6D**). For the urinary IgA, the total ELISA signal is lower than the urinary IgG, and SinH-123 vaccinated mice induced a statistically higher level of antigen-specific urinary IgA response than those in the control group when SinH-123 (GST-tag removed) was used as the ELISA antigen (Adjusted *P* value, *P* = 0.0258). When SinH-3 (GST-tag removed) was used as the ELISA antigen, SinH-3 vaccinated mice showed higher antigen-specific urinary IgA responses than the control group mice (Adjusted *P* value, *P* = 0.0024) (**Fig 6E**).

## SinH-based vaccination led to a minimal reduction in ExPEC colonization in healthy mice that was not statistically significant

The gastrointestinal tract is the major reservoir of ExPEC [83]. In addition, a recent study showed up to 93.5% of traditionally classified InPEC fecal isolates additionally carried ExPEC virulence factors, which might cause the infection outside of the GI tract [84]. Hence, the

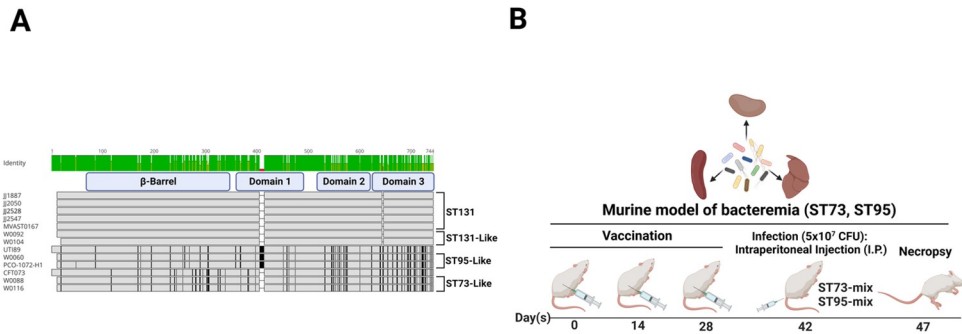

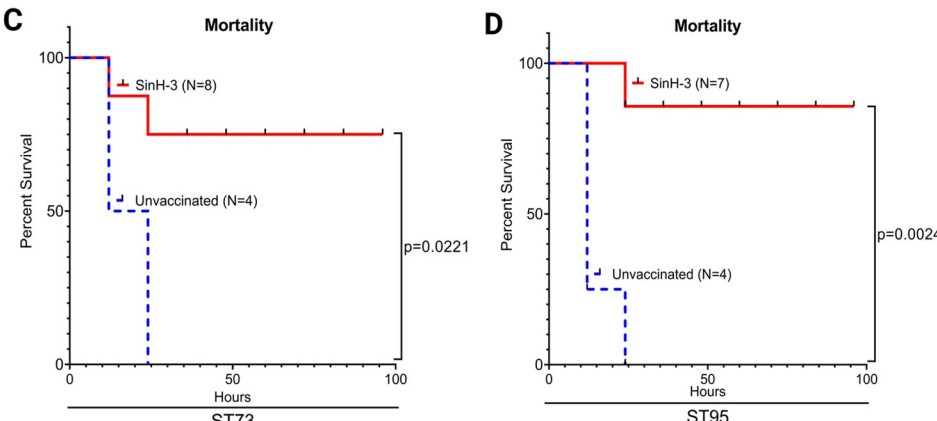

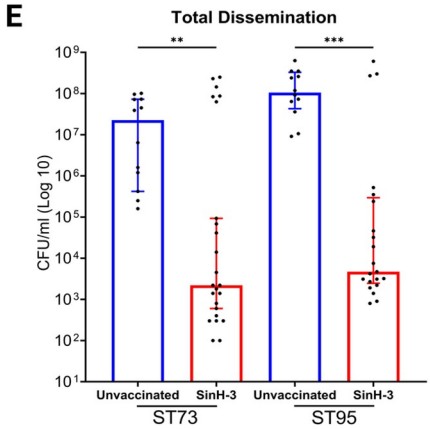

**Fig 5. Assessment of the protective efficacy of SinH-3 against the bacteremia of multiple ExPEC sequence types (STs).** (A) Sequence alignment of *sinH* in different sequence types of ExPEC. The alignment was exported from Geneious and annotated using BioRender. (B) The vaccination scheme was used in this experiment. BALB/cJ, 6 weeks old, unvaccinated female mice (N = 4) and female mice were subcutaneously immunized with SinH-3 (N = 8), were both injected with an intraperitoneal (IP) injection of $5 \times 10^7$ CFU of multiple ExPEC sequence type strains

(ST73-mixture, ST95-mixture). Mice were monitored twice a day for 5 days, and moribund animals were euthanized/ necropsied to determine bacterial levels in the kidneys, spleen, and liver. Organs were harvested and plated to determine bacteria levels. The schematic diagram was made in BioRender. The survival rate curve of (C) ST73-mixture or (D) ST95-mixture was determined using the Gehan-Breslow-Wilcoxon comparison. (E) Box-and-whisker plots of the bacterial levels (CFU/ml) of the counts from all organs following necropsy. Error bars indicate the median with 95% confidence interval (CI). Significant was determined by the Kruskal-Wallis analysis of variance (ANOVA) with Dunn's multiple comparisons correction. Symbols represent data of individual mice. One star (*) $P < 0.05$, two stars (**) $P < 0.01$, three stars (***) $P < 0.001$, four stars (****) $P < 0.0001$. The Box-and-whisker plots and Kaplan Meier survival curves were exported from Graphpad Prism 9 and annotated using BioRender.

reduction of the colonization of ExPEC or *E. coli* strains carrying ExPEC-associated virulence factors in this environment is expected to decrease the risk of extraintestinal infections throughout life. To test the hypothesis that SinH will reduce ExPEC colonization, mice were vaccinated as before and gavaged on day 42 with $10^9$ CFU of ExPEC ST131 strains JJ1886, JJ2547, JJ2050 (**Fig 7A**). Feces were homogenized, and the ExPEC bacterial load in the feces was quantified (**Fig 7B and 7C**). Although the reduction of bacterial loads was observed in the SinH-3 vaccinated group, the effect was not statistically significant (Adjusted *P* value, $P = 0.0797$) (**Fig 7B**). For the SinH-123 vaccinated group, there was no difference between the control group and the experimental group. In addition, SinH-3 vaccinated mice showed a better protective efficacy against ExPEC strains JJ2547 and JJ2050 colonization than the SinH-123 in the GI tract; however, the results were not statistically significant (**Fig 7C**).

## Immunization with SinH-based antigens reduced ExPEC colonization in the gastrointestinal tract in immunosuppressed mice

A significant proportion of *E. coli* bacteremia originates in immunosuppressed individuals, especially cancer patients receiving chemotherapy who are at high risk of developing neutropenia, which could severely decline the circulating immune cells. These patients usually suffer from long-term hospitalization and have relatively poor prognoses and high mortality rates [85]. To determine if SinH-based antigens are effective in the immunosuppressed clinical context to reduce the ExPEC colonization in the GI tract, we utilized a mouse model of chemotherapy-induced neutropenia, whereby immune cells of mice were damaged and declined by the injection of the chemotherapeutic agent cyclophosphamide [56,58]. Mice were vaccinated as before and were gavaged on day 42 with $10^9$ CFU of ExPEC ST131 strains (JJ1886, JJ2547, JJ2050), and the animals were intraperitoneally injected on alternate days with the cancer chemotherapy drug cyclophosphamide (Cytoxan [CTX]) on days 43, 45, 47 (**Fig 7D**). Feces samples were collected and homogenized on day 48, and the ExPEC bacterial load in the feces samples was quantified (**Fig 7E and 7F**). Although the results are not statistically significant, both SinH-3 and SinH-123 antigens showed a clear reduction in ExPEC colonization in the GI tract in immunosuppressed mice (Adjusted *P* value, SinH-3, $P = 0.0630$; SinH-123, $P = 0.0756$) (**Fig 7E**). In addition, both SinH-based vaccinations showed a better protective efficacy against ExPEC strain JJ2050 colonization in the GI tract in the immunosuppressed mice among three challenged ExPEC strains (Adjusted *P* value, SinH-3, $P = 0.0365$; SinH-123, $P = 0.0267$) (**Fig 7F**).

## Discussion

ExPEC is the leading cause of bacteremia and UTIs, persistent in the general community and hospitalized patients. Currently, this situation is exacerbated by overprescribing antibiotics, the spread of antibiotic-resistant plasmids, and the trend of global aging [86–88]. As a promising alternative strategy to combat this situation, developing an effective ExPEC vaccine to

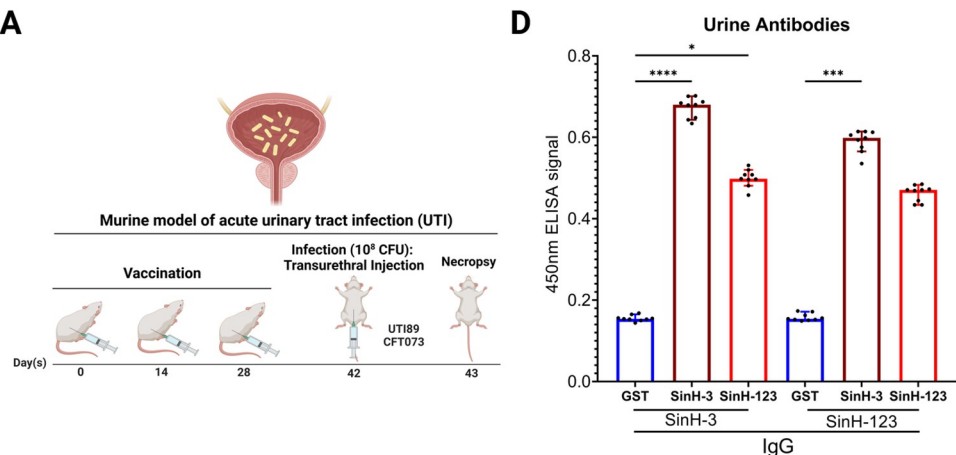

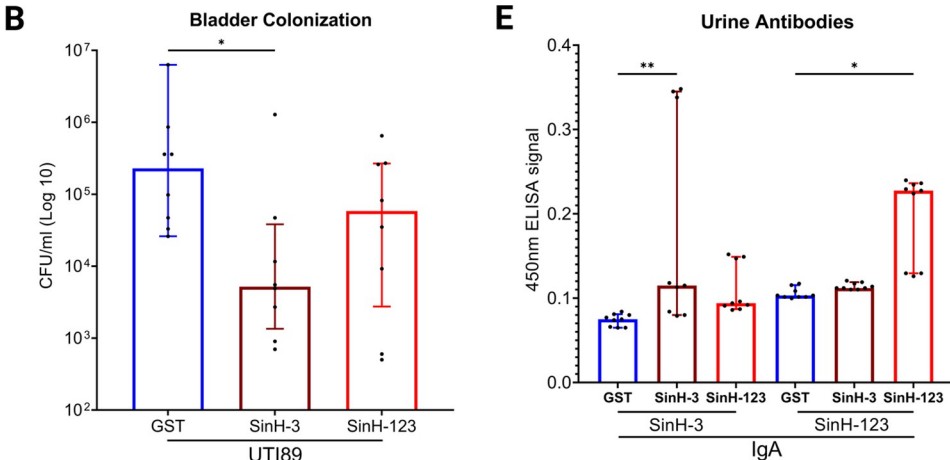

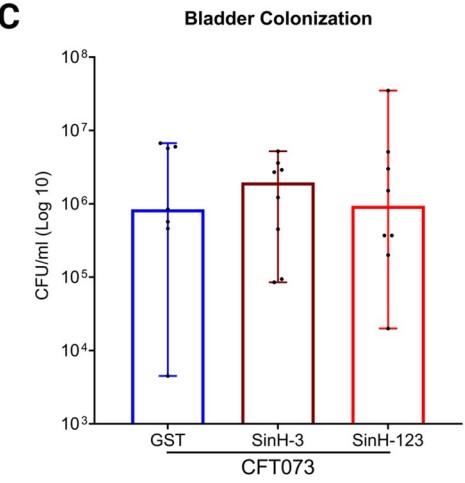

**Fig 6. Assessment of the protective efficacy and immunogenicity of SinH-based vaccines against acute urinary tract infection (UTI). (A)** The vaccination scheme was used in this experiment. BALB/cJ, 6 weeks old, female mice were subcutaneously immunized with SinH-based antigens (SinH-3, SinH-123, N = 8) or GST alone (N = 7 or 8) and

inoculated with a transurethral injection of $10^8$ CFU of UPEC strains (UTI89, CFT073). Bladders were harvested and plated to determine bacteria levels. Urine was taken from each mouse after complete immunization. The schematic diagram was made in BioRender. Box-and-whisker plots of the bacterial levels (CFU/ml) in the bladder of UTI89 **(B)** or CFT073 **(C)** ELISA analysis of urinary IgG **(D)** and IgA **(E)** from SinH-based antigens vaccinated animals using antigens, SinH-3 or SinH-123, as the capture antigen. Error bars indicate the median with 95% confidence interval (CI). Significant was determined by the Kruskal-Wallis analysis of variance (ANOVA) with Dunn's multiple comparisons correction. Symbols represent data of individual mice. One star (*) $P < 0.05$, two stars (**) $P < 0.01$, three stars (***) $P < 0.001$, four stars (****) $P < 0.0001$. The Box-and-whisker plots were exported from Graphpad Prism 9 and annotated using BioRender.

mitigate the increasing global burden of the AMR crisis and substantial public health burden would be tremendously beneficial to the population worldwide. Despite numerous attempts, no *E. coli* vaccine has been approved by the U.S. Food & Drug Administration (FDA). Here, we describe; (i) that immunization with either SinH-3 or SinH-123 reduced the bacterial burden of highly virulent ExPEC ST131 and increase the survival rate in the murine model of bacteremia; (ii) that immunization with SinH-based antigens produce a higher level vaccine-specific serum IgG, especially vaccination with SinH-123; (iii) that immunization with SinH-3 reduces UPEC strain UTI89 cystitis in the murine model of acute UTI; (iv) that immunization with SinH-based antigens produce a significantly higher level of vaccine-specific urine IgA and IgG; (v) that whereas immunization with SinH-based antigens did lead to a reduction in colonization compared to the control in both healthy and immunocompromised GI tract mice models, this was not statistically significant; (vi) that immunization with SinH-3 confers extensive protection against multiple ExPEC sequence types with different sinH-sequences in the murine model of bacteremia; (vii) that mice vaccinated with SinH-3 demonstrate a significant increase of survival rate after ExPEC (ST73, ST95) infection. In total, this data supports the contention that immunogens consisting of the extracellular domains of the autotransporter SinH represent promising ExPEC vaccine targets. To our knowledge, this is the first to report to utilize a pathogen-specific autotransporter protein as a ExPEC vaccine, an alternative to whole cell vaccines [20], O-antigen (ExPEC4V/ExPEC9V) conjugate vaccine [21–24], FimH vaccine [25–27], or iron acquisition receptor vaccine [31–36].

From the perspective of epidemiology, a vaccine against ExPEC would be expected to be effective against many of the 50 million incident cases of sepsis and 11 million sepsis-related deaths worldwide [89] and would contribute to the reduction of the UTI and recurrent UTI. For example, among all ages, both sexes, and all underlying causes, an estimated 41.5 million incidents of sepsis cases and 8·2 million sepsis-related deaths in 2017 occurred in countries and districts with a low, low-middle, or middle SDI (Socio-demographic Index (SDI) is a composite indicator of development status and strongly correlated with health outcomes). Countries with low, low-middle, or middle SDI would in particular benefit from an *E. coli* vaccine to reduce sepsis-related deaths and incidence [89]. Furthermore, bloodstream infections are the most frequent life-threatening infectious complication after abdominal solid organ transplantation, with morbidity and mortality rates up to 69% and 52%, respectively [90]. Recent reports have demonstrated that MDR gram-negative bacteremia has emerged frequently and become the predominant cause of morbidity and mortality after transplantation [91]. For this reason, an *E. coli* vaccine could be an important preventive strategy to reduce the incidence of post-transplant BSIs and control the spread of MDR organisms in the transplantation. In addition, from the age-related sepsis incidence perspective, overall, sepsis incidence peaked in early childhood, with a second peak in incidence among older adults. For example, in 2017, there were an estimated 20.3 million incident sepsis cases and 2·9 million deaths related to sepsis worldwide among children younger than 5 years. When applied clinically, a SinH-based vaccine could be used to vaccinate children younger than 5 years and older adults (over 50 years)

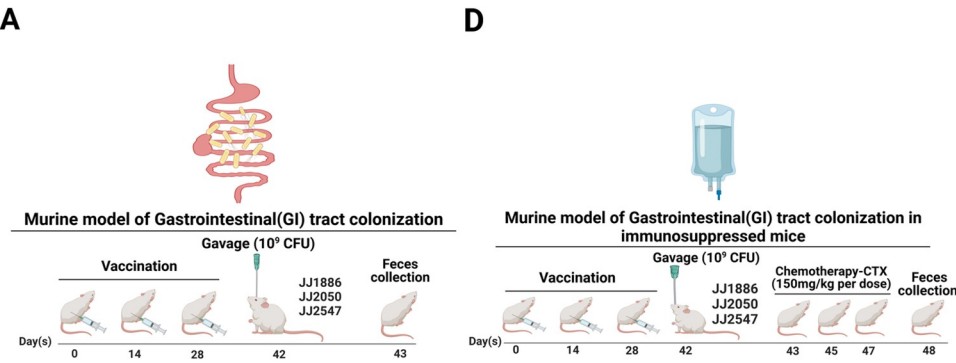

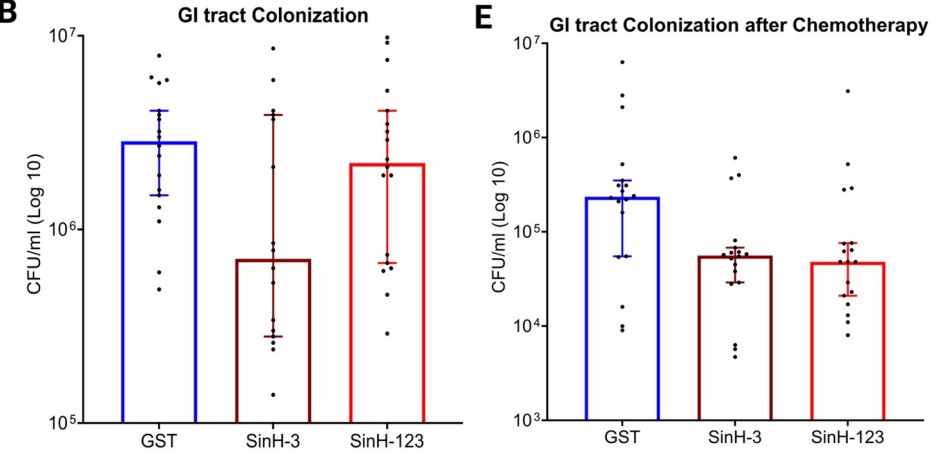

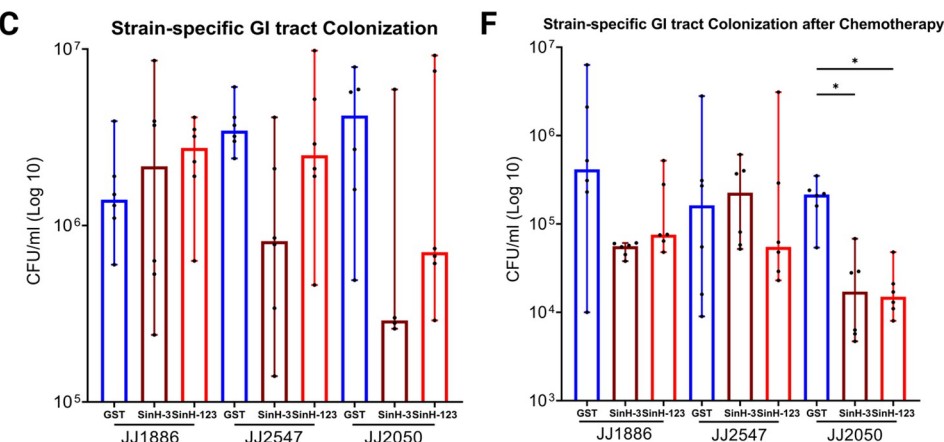

**Fig 7. Assessment of the protective efficacy of SinH-based vaccines against ExPEC colonization in the GI tract.**
(**A**) The vaccination scheme was used in the murine model of gastrointestinal (GI) tract colonization. BALB/c, 6 weeks old, female mice were subcutaneously immunized with SinH-based antigens (SinH-3, SinH-123, N = 18) or GST alone

(N = 18) and inoculated with a gavage of $10^9$ CFU of ExPEC ST131 strains (JJ1886, JJ2547, JJ2050). Feces samples were collected and plated to determine bacteria levels. **(B)** Box-and-whisker plots of the bacterial levels (CFU/ml) in combining the counts from all ExPEC strains (JJ1886, JJ2547, JJ2050) **(C)** or the bacterial levels (CFU/ml) of each ExPEC strain in feces. **(D)** The vaccination scheme was used in the murine model of gastrointestinal (GI) tract colonization in immunosuppressed mice. BALB/c, 6 weeks old, female mice were subcutaneously immunized with SinH-based antigens (SinH-3, SinH-123, N = 18) or GST alone (N = 18) and inoculated with a gavage of $10^9$ CFU of ExPEC ST131 strains (JJ1886, JJ2547, JJ2050). And then, mice were treated with the chemotherapeutic agent Cytoxan (CTX) on alternate days. After three times injections, feces were harvested and plated to determine bacteria levels in immunosuppressed mice. **(E)** Box-and-whisker plots of the bacterial levels (CFU/ml) in combining the counts from all ExPEC strains (JJ1886, JJ2547, JJ2050) **(F)** or the bacterial levels (CFU/ml) of each ExPEC strain in immunosuppressed mice feces. Error bars indicate the median with 95% confidence interval (CI). Significant was determined by the Kruskal-Wallis analysis of variance (ANOVA) with Dunn's multiple comparisons correction. Symbols represent data of individual mice. One star (\*) $P < 0.05$, two stars (\*\*) $P < 0.01$, three stars (\*\*\*) $P < 0.001$, four stars (\*\*\*\*) $P < 0.0001$. The schematic diagrams were made in BioRender. The Box-and-whisker plots were exported from Graphpad Prism 9 and annotated using BioRender.

to reduce the sepsis incidence and death rate among this population [89]. Meanwhile, UTIs are the most common outpatient infections, with a lifetime incidence of 50–60% in adult women, especially women over 50 years of age. Thus, the vaccine proposed here could be used to protect this at-risk population.

Although the SinH-based antigens demonstrated high-efficiency protection in the murine model of bacteremia against multiple ExPEC sequence types of colonization, the protective efficacy of SinH-based antigens is not as adequate in the acute UTI model and acute GI tract model as expected. One potential reason for efficacy is that the virulence functions of autotransporter proteins include adhesion, aggregation, and invasion [92]. It is hypothesized here that bacterial clearance is simultaneously mediated by opsonization (opsonophagocytosis), neutralization, and other functions of the antibodies which may either block SinH function (prevent adherence or invasion) or, since its surface-localized, induce its uptake by macrophages. Also, urinary IgG demonstrates a greater level of protection against ExPEC colonization in the urinary tract than urinary IgA, which indicates the high level of urinary IgG is still essential in this mucosal site. In addition, possible differences in the abundance or exposure of SinH on the bacterial surface may explain the observed differences in vaccines efficacy against UTI89 and CFT073 in the murine model of acute UTI. Another potential reason is the deficiency of colonization time post-inoculation. For achieving the acute urinary tract infection and GI tract infection, we only allowed the infections to last 24 hours. However, unlike the intraperitoneal injection, in which bacteria were absorbed from the peritoneal cavity by the portal system with faster speed, transurethral and gavage inoculation would allow the bacteria to colonize on the mucosal site. Hence, SinH-based antigens vaccinated mice might provide a more mucosal immune response and protection against ExPEC colonization in the urinary tract and GI tract if increasing the colonization time after the inoculation until 48 hours or more.

Variations in the immunization route, different adjuvant and mouse model strain all impact the evaluation of vaccine efficacy. Mucosal immunization could efficiently induce local immune responses to pathogens at mucosal sites and efficiently generate immune responses detectable at distant mucosal tissues and in the blood [93]. In addition, previous studies indicated mice intranasally immunized with the iron receptor, FyuA, elicited a long-term vaccine-specific humoral immune response, and reduced the UPEC kidney colonization after transurethral challenged with $10^8$ CFU of UPEC strain 536 [32]. Another study showed intranasal immunization with iron-containing receptors, such as Hma, IreA, or IutA generates an antigen-specific humoral response and antigen-specific IL-17 and IFN-γ; also, mice immunized with the IreA have significantly reduced the CFT073 bacterial counts in the bladder [31]. Hence, without the impaction of the high protective efficacy of SinH-based vaccines in the blood, intranasal or transurethral immunization might be an alternative way to increase the

protective efficacy of vaccinated mice against ExPEC colonization in both urinary tract infection and other organs.

Furthermore, without the modification of subcutaneous immunization, immune responses induced by vaccines might be drastically enhanced with the use of other adjuvants. In this study, the adjuvant that we used is alum, which would enhance the immune response by facilitating phagocytosis and accumulating the inflammatory cells. Although alum has been recorded as excellent in safety and the most used adjuvant on a 70-year history of use, it does not elicit as strong an immune response as other adjuvants [94]. In preclinical models, the most largely utilized adjuvants to induce mucosal immune responses have been non-toxic derivatives of cholera toxin [95]. Some studies showed unlike other Toll-like receptors (TLR)-based adjuvants, the adjuvant dmLT induces strong IL-17 cytokine secretion and antigen-specific Th17 responses after parenteral or mucosal immunization, which is critical in protection from pathogens [96]. In addition, the dmLT adjuvant has been shown to enhance mucosal responses to the oral inactivated enterotoxigenic *Escherichia coli* (ETEC) vaccine ETVAX by increasing the production and secretion of mucosal IgA antibodies and inducing IL-1β as well as other cytokines [97]. Hence, dmLT might be a potential alternative adjuvant to enhance the mucosal immune response and vaccine-specific urinary IgA of the SinH-based vaccines. A recent study demonstrated that UTIs typically evoke prompt and vigorous innate bladder immune responses, including extensive exfoliation of the epithelium; however, following each bladder infection, a highly T-helper type 2 (Th2) immune response would preferentially repair the bladder epithelial cells, which would proportionally inhibit T-helper type 1 (Th1)-mediated responses, especially those related to bacteria-clearing activities, resulted in the reinfections [98]. Therefore, the adjuvant oligodeoxynucleotides containing unmethylated CpG (CpG ODNs), mounting an innate immune response characterized by the generation of Th1 and pro-inflammatory cytokines with a good safety profile in clinical trials [99], could also be an alternative adjuvant in SinH-based vaccines, which could enhance the Th1-mediated bacteria-clearing responses to balance the Th2 re-epithelialization in the mucosal sites of both the urinary tract and GI tract, and also might increase the vaccine-specific serum IgG level in the blood.

Under the global AMR crisis, a SinH-based vaccine may contribute to a promising alternative strategy to combat the increasing global burden of the AMR, effectively mitigating the expansion of resistance elements. In addition, by bridging computational genomics with virulome vaccinology, a similar approach may be utilized for other genetically pleiotropic bacteria such as MDR *Klebsiella pneumoniae* and *Staphylococcus aureus*.

## Supporting information

**S1 Fig. Determination of the putative GST-SinH-3 and GST-SinH-123 proteins by Mass Spectrometry per-band sequencing.** Purified protein bands were resolved and digested in gel. The tryptic peptides were analyzed on nanospray LC-MS (liquid chromatography-mass spectrometry) system. The eluted peptides were directly electro-sprayed into mass spectrometer and analyzed by data-dependent acquisition (DDA). The coverage (the percentage of the protein sequence by identified peptides) of GST-SinH-3 was shown in (A), and the coverage of GST-SinH-123 was shown in (B).
(DOCX)

## Acknowledgments

We thank James R. Johnson for allowing us to use ExPEC strains JJ2050, JJ1886, and JJ2547 from his collection. BCM Mass Spectrometry Proteomics Core is supported by the Dan L.

Duncan Comprehensive Cancer Center NIH award and CPRIT Core Facility Award. We thank Professor Robert A. Britton, Professor Pedro A. Piedra, and Professor Pablo C. Okhuysen for their input. We also thank Ellen Vaughan, Hannah Carter, Carmen Gu Liu, Keiko Salazar, and Austen Terwilliger for discussing information on certain virulence factors in this study.

## Author Contributions

**Conceptualization:** Yikun Xing, Anthony W. Maresso.

**Data curation:** Justin R. Clark.

**Formal analysis:** James D. Chang.

**Funding acquisition:** Anthony W. Maresso.

**Investigation:** Yikun Xing, Anthony W. Maresso.

**Methodology:** Dylan M. Chirman, Sabrina Green, Jacob J. Zulk, Joseph Jelinski, Kathryn A. Patras.

**Software:** Justin R. Clark, James D. Chang.

**Writing – original draft:** Yikun Xing.

**Writing – review & editing:** Yikun Xing, Justin R. Clark, James D. Chang, Kathryn A. Patras, Anthony W. Maresso.

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
