## [Decision Letter · Decision Letter 0]

29 Jun 2022

Dear Dr. Maresso,

Thank you very much for submitting your manuscript "Comparative genomics to determine a vaccine antigen - Broad Protective Vaccination against Systemic Escherichia coli with Autotransporter Antigens" for consideration at PLOS Pathogens. As with all papers reviewed by the journal, your manuscript was reviewed by members of the editorial board and by several independent reviewers. In light of the reviews (below this email), we would like to invite the resubmission of a significantly-revised version that takes into account the reviewers' comments.

We cannot make any decision about publication until we have seen the revised manuscript and your response to the reviewers' comments. Your revised manuscript is also likely to be sent to reviewers for further evaluation.

Sincerely,

Kimberly Kline

Pearls Editor

PLOS Pathogens

Christoph Tang

Section Editor

PLOS Pathogens

Kasturi Haldar

Editor-in-Chief

PLOS Pathogens

orcid.org/0000-0001-5065-158X

Michael Malim

Editor-in-Chief

PLOS Pathogens

orcid.org/0000-0002-7699-2064

Reviewer's Responses to Questions

**Part I - Summary**

Reviewer #1: Xing and colleagues describe the use of an E. coli autotranporter with homology to Yersinia invasin, SinH, for subcutaneous vaccination of mice, with Alum as adjuvant to protect against ExPEC ST131 strains (and others). The authors have previously shown that sinH has high prevalence in certain ExPEC phylogroups. Two domains of the protein were expressed as GST fusions and partially purified prior to use in their vaccine prep in intraperitoneal model of bacteremia, transurethral challenge of the mouse bladder for urinary tract infection and oral gavage for gastrointestinal colonization. Statistically significant reduction in colonization was observed for bacteremia and for intestinal colonization but not bladder colonization. Antibodies to proteins in the vaccine prep were elevated following vaccination and dramatically improved survival of vaccinated mice. While there have been many attempts to develop and ExPEC vaccine, none have yet been licensed. However, there has been a lot of work in the area. Nevertheless, it is important to keep working on vaccines until a suitable candidate can be found. Indeed, SinH may be such a viable candidate.

Reviewer #2: This manuscript by Xing and coauthors presents an extensive study seeking a vaccine target for systemic E. coli infections. The investigators used a comparative genomics approach to identify a sinH as selectively present in ExPEC, identify it as encoding a plausibly surface-exposed protein amenable to antibody recognition in a live bacterium, use structure prediction to identify a ligand binding domain, immunize mice to two versions of this domain derived from the multidrug resistant ST131, and evaluate outcomes in three distinctive infection/colonization models using different E. coli strains.

The authors provide compelling evidence of protection against mortality for two strains in a mouse septicemia model with a corresponding difference in multiorgan CFU values. A relatively marginal (though narrowly significant) CFU decrease in GI colonization and bladder CFUs with a UTI model are presented. Results are considered in the context of antibody responses. Similar efforts, using different antigens have been attempted by other groups and these efforts are appropriately noted in this manuscript, where they provide informative context.

A strength of this manuscript is its principled scientific approach to vaccine development, which begins with a thoughtful discovery approach in ExPEC-associated phyla and sequence types, considers protein structure, and evaluates results in more than one E. coli strain using GST tag immunization as a negative control. The manuscript would be improved by equivalent consideration of ST131 and non-ST131 strains in the septicemia model, particularly regarding mortality, given the focus upon, and use of, an ST131 vaccine antigen. While the vaccine as evaluated here did not appear to yield the desired pan-isolate protective effect that was intended, the differences between strains and mouse models (sepsis, UTI, gut colonization) are interesting and may prove to be instructive for future work. The manuscript would be improved if the presentation of these limitations was presented in a more direct manner.

Reviewer #3: This manuscript evaluated the protective efficacy and immunogenicity of a recombinant SinH-based vaccine against infections caused by extraintestinal E. coli.

**Part II – Major Issues: Key Experiments Required for Acceptance**

Reviewer #1: 1. The abstract (line 28) and results (line 121) call sinH an uncharacterized gene. However, when searching PubMed for “sinH and E. coli”, four references are identified. One is the genomic sequence screens described previously by the senior author’s group. The most recent publication, however has characterized the sinH gene and its respective predicted protein domains based on protein prediction software. According to the abstract, he gene was mutated and investigated in the murine urinary tract infection model. The abstract also states that a sinH mutant has a fitness defect and its gene is upregulated during urinary tract infections in women. This is a 2022 paper and was likely missed during manuscript preparation, but appears to provide important background that should now be included.

Shea AE et al. Loss of an Intimin-Like Protein Encoded on a Uropathogenic E. coli Pathogenicity Island Reduces Inflammation and Affects Interactions with the Urothelium. Infect Immun. 2022. 90:e0027521.

2. Citations are somewhat spotty. For example when reviewing antigens that have been tried as vaccine for ExPEC, perhaps the most promising one, FimH of type 1 fimbria is not mentioned. Siderophores were also tried. Other references do not always cite the primary literature. Sometimes reviews are not the most relevant articles to cite. Some are omitted altogether. A more thorough assembly of these references should be added. Also (beginning line 135), primary references should be provided for published strains. Also, unless the authors developed the three murine models in this study, primary references should be given for those models in the Methods sections . As well, the vaccination protocol should be referenced unless developed by the authors

3. Some figures are illegible or uninterpretable as downloaded. For example Fig 1 B and C. and Fig 4A.

4. With respect to the “purified antigens”, one can see the predicted induced bands and when cut out and subjected to mass spec, they are clearly correct, but there are many many other bands in the a-little-hard-to-see Fig 2F (perhaps incompletely destained). Presumably this is what was used for vaccination. It cannot be ruled out that these other bands may be what is affording protection and to what the ELISA is reacting. The authors should show an SDS-PAGE of what was used for vaccination or a Western blot showing reactivity to only the SinH domains. Furthermore, a vaccination trial that has only alum, only sinH and the combo (the latter done) should be done. Challenging with a sinH mutant of the ST131 strain should demonstrate no protection since it lacks the key antigen. This represents a key experiment to clarify the protection data and to demonstrate SinH domains are actually protectinve and not contaminants.

Reviewer #2: The mouse models consider three rather distinctive circumstances: 1) septicemia from experimental peritonitis; 2) urinary tract infection – specifically cystitis/bladder infection; 3) gastrointestinal colonization. Each of these is associated with very different clinical contexts, immunologic responses, microenvironments, and bacterial functions. The most compelling result presented here is the vaccine mortality benefit (presented in Figure 4) observed in the septicemia model of ST73 and ST95 infections, which is corroborated by a substantial difference in organ CFUs. Curiously missing from this paper are survival results for the ST131 strains presented before this in Figure 3. I see no report of mortality data for these strains in the corresponding Results section (lines 403-442). This was surprising as the SinH-3 domain from ST131 (lines 142-147) differs by 30 amino acids from the same proteins in ST73 and ST95. I see a less marked vaccine-associated CFU drop for ST131 strains compared to the non-ST131 strains. Does ST131 express less antigen? Did the ST131 strains somehow evade humoral immunity to a greater degree than ST73/95? Given that the authors focused upon ST131 strains and septicemia, this is a very important result that merits more direct treatment. Please provide the data for ST131 survival so it may be directly compared to the survival data for the non-ST131 strains.

Reviewer #3: L298-335. The authors claim the sequence of SinH from ST131 is significantly divergent compared to the SinH sequence from ST73 and ST95. What is the level of divergence? My quick analysis indicated the SinH protein sequence identity between reference ST131 strains compared to reference ST73 and ST95 strains is ~90% identity. Following on from this, I found Fig 1 very difficult to interpret. In panel A, black boxes indicate ‘no results’ – what does this mean? I could not resolve any information from panel B and panel C as the text was too small.

The SinH-3 and SinH-123 recombinant proteins were not pure - the SDS-PAGE analysis for both protein preparations shows multiple bands, including a significant band at ~25 kDa (see Fig 2F). The authors need to determine if these bands are breakdown products or impurities. They should also determine the purity of the sample using mass spectrometry and demonstrate there is no LPS contamination. These experiments are critical, as the authors need to demonstrate that the reduced cfu counts resulting from vaccination are due to an antibody response against SinH rather than against other ExPEC proteins (or LPS) co-purified in the procedure.

It would be relevant to employ the 5-day IP infection protocol to assess vaccine protection and survival (the critical readout) with the ST131 strains. Please comment.

Is the data showing protection against colonisation of the bladder by UTI89 from a single experiment? If so, this experiment should be repeated to demonstrate reproducibility.

Likewise, is the gut colonisation data from a single experiment?

I have concerns about the statistical methods used to evaluate the significant difference in cfu counts in different experiments. Data in Fig 3 and Fig 5 were evaluated using a two-tailed Mann-Whitney test or Kruskal-Wallis test (which one is not specified in the figure legend), while data in Fig 4 and Fig 6 were evaluated using a two-tailed Mann-Whitney test. Please explain. Given there are multiple comparisons, I would have expected these analyses would be performed using a test such as the Kruskal-Wallis analysis of variance (ANOVA) with Dunn’s multiple comparisons correction. Please justify the choice of statistical method.

**Part III – Minor Issues: Editorial and Data Presentation Modifications**

Reviewer #1: 5. In Fig 4E, vaccinated with SinH-3 and challenged with ST73, there are six data points (>25% of mice) that are nearly higher than any control value. Please check the statistics on this. The stats would likely take a big hit with these values

6. Some awkward and incorrect language:

Line 151 ‘..proteins were cultured in E. coli BL21(DE3)”. The proteins were not cultured, the E. coli was.

Line 148. plasmid

7. Line 152 “the proteins were induced”. Gene expression was induced…

8. Line 157 should “PSIG” just be “PSI”?

9. Lines 217, 219, 225, 227 (for example) Throughout the manuscript “infection” is used instead of “injected” or “inoculated”. The animals may become infected but only some time after injection or inoculation.

10. Line 391. I don’t think the SinH-based antigens were harvested by centrifugation. Likely the insoluble bacterial material was pelleted. The supernatant likely contained the antigens.

11. Graph in Fig 3E should read” Unvaccinated” on x-axis.

12. In Fig 5B, there is no trend in this experiment. Simply state that the differences were not significant. One cannot put such a spin on these data. Likewise for Figure 6B (SinH-123).

Reviewer #2: 1) Assuming equivalent antigen expression (not examined in this manuscript), among the three models tested (sepsis, UTI, intestinal colonization), I would expect the humoral immune response to subcutaneous vaccination to have the greatest impact upon septicemia because of circulating antibodies, as is found here. Since these invasive infections are what kill patients, I regard this as an important, if not the most important, outcome of the current study. I am concerned that the current manuscript understates this distinction (lines 32-35, for example). The survival data in Fig 4 provides the clearest example of protection, while the meaning of the relatively narrow CFU differences associated with other organisms and contexts is more difficult to discern. While perhaps an imperfect analogy, the concept that a subcutaneous vaccine could confer greater protection against an invasive disease than to a pathogen confined to the mucosa has become highly familiar in the age of COVID vaccines.

2) Unmentioned in this manuscript is the subject of sinH transcription and expression as a modulating influence on the results. What is known about this? Are there any clues from the sinH promoter region?

3) Line 552: I agree with the authors’ suggestion that Fc-mediated effects that are not active at mucosal surfaces or intestinal/bladder lumen is a potential explanation for the diminished effect of vaccination on colonization. Can the authors comment on the likelihood that antibodies that bind SinH would affect SinH-mediated bacterial functions? Has anybody examined virulence of a SinH knockout? On a related note, Line 556: urinary IgG didn’t “demonstrate more effective protection” in this study, rather was associated with greater protection.

Reviewer #3: L410-411. Please define the genetically distinct clonal groups for JJ1886, JJ2050 and JJ2547.

Were the mice sick at 24 hours post infection with the ST131 strains? The data for the ST73 and ST95 strains would indicate that many of the mice may have died within this time (i.e. compare with survival data in Fig 4).

Please refer to strains by their strain name rather than ST. As noted by the authors, there is considerable variation at the ExPEC genome level, and thus while the range of strains examined is to be commended, single strains are not representative of an entire ST.

In the discussion, it is stated that ‘from the perspective of epidemiology, a vaccine against ExPEC would be expected to be effective against many of the 50 million incident cases of sepsis and 11 million sepsis-related deaths worldwide and would contribute to the reduction of the recurrence rate of complicated UTI. The authors should comment on a how a vaccine for E. coli would be used. How will they identify the high-risk target group? Surely it would not be administered to everyone.

Although SinH is immunogenic, the results actually show it is not a protective antigen. For example, at 24h bacteria were still present in the blood of all mice (Fig 3). Even more concerning from a clinical perspective, the challenge strain was present in the spleen and kidney in high numbers in immunized mice (albeit at lower cfu numbers than the control), indicating a lack of protection in these tissues. The authors should comment.

Do the authors envisage SinH would be used as part of a multivalent vaccine? Some discussion is warranted.

PLOS authors have the option to publish the peer review history of their article (what does this mean?). If published, this will include your full peer review and any attached files.

Reviewer #1: No

Reviewer #2: No

Reviewer #3: No
---

## [Decision Letter · Decision Letter 1]

28 Nov 2022

Dear Dr. Maresso,

Thank you very much for submitting your manuscript "Broad Protective Vaccination against Systemic Escherichia coli with Autotransporter Antigens" for consideration at PLOS Pathogens. As with all papers reviewed by the journal, your manuscript was reviewed by members of the editorial board and by several independent reviewers. The reviewers appreciated the attention to an important topic. Based on the reviews, we are likely to accept this manuscript for publication, providing that you modify the manuscript according to the review recommendations.

Sincerely,

Kimberly A. Kline

Pearls Editor

PLOS Pathogens

Christoph Tang

Section Editor

PLOS Pathogens

Kasturi Haldar

Editor-in-Chief

PLOS Pathogens

orcid.org/0000-0001-5065-158X

Michael Malim

Editor-in-Chief

PLOS Pathogens

orcid.org/0000-0002-7699-2064

Reviewer Comments (if any, and for reference):

Reviewer's Responses to Questions

**Part I - Summary**

Reviewer #1: see my original review. The revised manuscript will be an important and much improved contribution to the literature

Reviewer #2: This revised manuscript describes broad spectrum protection from septicemic death following vaccination with an autotransporter protein of unknown function. The results are of interest in the long term effort, extending over many years, to develop an E. coli vaccine protective against extraintestinal infection. In that context, principled investigations of new antigens such as this are of interest regardless of experimental results. Both the successes and limitations of this work are worth careful consideration to guide and evaluate future work. Here, notable protection against septicemia is observed, while the effects of vaccination in the animal models of cystitis and intestinal colonization are relatively marginal. The results at these mucosal surfaces are relatively typical of vaccination strategies to date, despite the categorical change in antigen.

Why results vary depending upon the infecting strain evaluated is curious and remains unaddressed.

Revisions have improved the manuscript.

Reviewer #3: (No Response)

**Part II – Major Issues: Key Experiments Required for Acceptance**

Reviewer #1: The authors have bent over backwards to answer all critiques of all three reviewers. The manuscript is greatly improved

Reviewer #2: (No Response)

Reviewer #3: (No Response)

**Part III – Minor Issues: Editorial and Data Presentation Modifications**

Reviewer #1: none

Reviewer #2: 1) Lines 583-598: The data in this section is explicitly characterized by the subheading as demonstrating that GI tract colonization is reduced by immunization (this is also a conclusion in Discussion lines 632-4), yet none of the results for this are significant and the effect sizes are small. The data as presented does not support the stated conclusion. Immunization has a non-significant, minimal effect on GI colonization by ExPEC strains.

2) Lines 520-526: given the high mortality of the infection, are the bacterial levels measured here only from survivors? This should be stated more clearly in this subsection. Would not be surprised to see lower CFU in survivors and agree it is worth presenting these data as explicitly confirmatory of the survival data. A summary of the data in this subsection in which mortality and CFU analyses are related would be useful for readers.

3) Consider restatements of the following:

- Line 104-106: I don’t think this was the authors’ interest but the suggestion that a beta lactamase has fueled fluoroquinolone resistance sounds like a misunderstanding of beta-lactamase function, which would not be expected to directly affect fluoroquinolone susceptibility.

- Line 106-107: “…unlike the other ExPEC sequence type, ST131 has more extensive genome content and an expanded number of virulence genes…”. This is a misleading statement, as the results in the cited review come from a comparative genomic study to a specific set of antibiotic-resistant ExPEC sequence types. Unlike that comparison, the present study compares ST131 to the more common ExPEC ST73, 95, and 127. It may be that the identity and combination of virulence genes, not the number of them, is determinative for ST131 pathogenicity and transmission. This is not a critical point to the manuscript but is a detail worth clarifying.

- Is the seventh conclusion in lines 636-637 the same as the first conclusion in lines 627-628? This seems like the primary conclusion stated once more.

- Lines 645-646: “a vaccine against ExPEC… would contribute to the reduction of the recurrence rate of complicated UTI”. Singling out “complicated UTI”, which is a clinical subcategory of UTI, seems odd here. Why not just say it would be effective against UTI or recurrent UTI?

Reviewer #3: This revised manuscript is significantly improved. I have the following additional comments.

L106-7. This is incorrect. ST131 does not have more extensive genome content and an expanded number of virulence genes compared to other ExPEC.

L115-116. I am not aware that O antigen vaccines exhibit weak immunogenicity.

L585-587. I don’t understand this statement. Please reword for clarity. ‘In addition, in fecal E. coli strains carrying ExPEC-associated virulence factors, up to 93.5% of fecal isolates classified as InPEC additionally carried ExPEC virulence factors’

P583-598. The gut colonisation experiments are confusing. In the methods this is written as one experiment with 2 parts. However, in the results it is written as 2 independent experiments, and this is also depicted in Fig 7A and 7D. Please clarify.

Why were cfu’s only determined from feces at one timepoint? If more data is available, this should be shown, even if the differences are not significant between the groups.

L632-4. This statement is not consistent with the data. Please correct. ‘(v) that immunization with SinH-based antigens are effective at reducing the colonization of ExPEC ST131 in the GI tract of healthy or immunocompromised mice’

PLOS authors have the option to publish the peer review history of their article (what does this mean?). If published, this will include your full peer review and any attached files.

Reviewer #1: No

Reviewer #2: No

Reviewer #3: No

Figure Files:

Data Requirements:

Reproducibility:

References:

---

## [Editor Report · Decision Letter 2]

26 Dec 2022

Dear Dr. Maresso,

We are pleased to inform you that your manuscript 'Broad Protective Vaccination against Systemic Escherichia coli with Autotransporter Antigens' has been provisionally accepted for publication in PLOS Pathogens.

Best regards,

Kimberly A. Kline

Pearls Editor

PLOS Pathogens

Christoph Tang

Section Editor

PLOS Pathogens

Kasturi Haldar

Editor-in-Chief

PLOS Pathogens

orcid.org/0000-0001-5065-158X

Michael Malim

Editor-in-Chief

PLOS Pathogens

orcid.org/0000-0002-7699-2064
---

## [Editor Report · Acceptance letter]

3 Feb 2023

Dear Dr. Maresso,

We are delighted to inform you that your manuscript, "Broad Protective Vaccination against Systemic Escherichia coli with Autotransporter Antigens," has been formally accepted for publication in PLOS Pathogens.

Best regards,

Kasturi Haldar

Editor-in-Chief

PLOS Pathogens

orcid.org/0000-0001-5065-158X

Michael Malim

Editor-in-Chief

PLOS Pathogens

orcid.org/0000-0002-7699-2064